

# Do composted bioamendments enhance the resistance of Mediterranean agricultural soils and their microbial carbon use efficiency to extreme heat-stress events?

Sana Boubehziz[1α], Emily C. Cooledge[2α], David R. Chadwick[2], Vidal Barrón[1],
Antonio Rafael Sánchez-Rodríguez[1], Davey L. Jones[2]

[α] These authors contributed equally to the manuscript and are considered co-first authors.

[1] *Department of Agronomy, Universidad de Córdoba, 14071, Córdoba, Spain*

[2] *School of Environmental and Natural Sciences, Bangor University, Gwynedd, LL57 2UW, UK*

*Corresponding to: Sana Boubehziz (z12boubs@uco.es)*



**Abstract.** Mediterranean agroecosystems are vulnerable to extreme heat-stress, especially because of their low organic matter content. Bioamendments may enhance soil nutrient content and microbial resilience to heatwaves. However, their effectiveness under these conditions is still unclear. We investigated the effect of bioamendments (composted olive mill pomace, biosolids and solid urban residue) and a conventional fertiliser (diammonium phosphate) on microbial carbon use efficiency (CUE), and soil biogeochemistry in two different soils, a calcareous

Vertisol and a non-calcareous Inceptisol, with low P availability, subjected to extreme heat-stress. We conducted warming experiments (20, 30, 40, or 50 °C), to monitor $^{14}$C-glucose mineralization and to evaluate modifications in soil biochemical properties. As result of warming, both soils microorganisms exhibited thermotolerance until 40 °C, with a critical shift in microbial respiration observed at 50 °C. Consequently, microbial CUE, which was a function of the bioamendments and soil, significantly declined from 0.47-0.65 at 20 °C to 0.27-0.45 at 50 °C ($p <$

0.05), with the control decreasing by $0.010 \pm 0.001$ °C$^{-1}$ (Vertisol) and $0.007 \pm 0.001$ °C$^{-1}$ (Inceptisol). Moreover, composted olive mill pomace-treated soils enhanced the resistance of soils to heat stress as they produced the highest microbial CUE at 40 °C in the Inceptisol and 50 °C in both soils ($0.43 \pm 0.02$ Inceptisol vs. $0.45 \pm 0.02$ Vertisol). Soil biogeochemistry varied with temperature and treatment, while available P in soils treated with diammonium phosphate was reduced with temperature in both soils, and available P added with bioamendments

was not affected by temperature but was increased with biosolids for all temperatures in the Inceptisol. In conclusion, organic matter rich bioamendments (composted olive mill pomace) may enhance the resistance of Mediterranean agricultural soils subjected to extreme heat-stress events (50 °C).


**Keywords**

Heatwave, organic matter, olive mill pomace, microbial respiration, phosphorus.



## 1. Introduction

Extreme weather events, such as heatwaves, and droughts in Mediterranean regions have become more severe in recent years (Wedler et al., 2023). Summer temperatures in the western Mediterranean region have reached high records and are expected to become more intense over the next few decades (Tejedor et al., 2024). The increase of global warming affects soil microorganisms and their activity in this region (Bañeras et al., 2022; Bérard et al., 2011), which involves changes in their structure and functioning, hence, altering nutrient cycling at a regional and global scale (Frey et al., 2013; Mooshammer et al., 2017; Reichstein et al., 2013). This is predicted to have a negative impact on soil functionality (e.g., soil carbon (C) sequestration and nutrient cycling), and lead to harmful environmental (Harris et al., 2018; Qu et al., 2024) and socioeconomic (Sun et al., 2024; Yin et al., 2022) losses across the agrifood system. As such, by 2050 it is estimated that the combined impact of heat-stress and drought will reduce global food production by 6-14 %, placing 0.56-1.36 billion people at risk of food insecurity (Kompas et al., 2024), losses in labour productivity due to heat stress are expected to reduce global gross domestic product by 2.5 % (Sun et al., 2024).

Despite intergovernmental efforts to limit global warming to 2 °C (Dosio et al., 2018), the frequency, intensity, and duration of heatwaves continues to increase (Seneviratne et al., 2021), further exacerbating soil desiccation in high-risk semi-arid regions of southern Europe and the Mediterranean area (Diffenbaugh et al., 2007; Fischer and Schär, 2010; García-García et al., 2023; Perkins-Kirkpatrick and Lewis, 2020). The Mediterranean region is one of the most exposed in the world to catastrophes, drought, biodiversity loss and land degradation (Antonelli et al., 2022). Calcareous soils are common in these areas, and they are characterised by poor organic matter content, which affects soil functionality and quality, and limited P and micronutrient availability (due to their pH being carbonate buffered to values above 8 because of their carbonates content), often leading to a negative impact on soil fertility and plant productivity (Helyar et al., 1974; Singh and Dahiya, 1976). In addition, dry conditions in these regions can amplify heatwave intensity and enable topsoil temperature to reach up to 50 °C during an extreme heat-stress event (Hamdi et al., 2011; Perkins, 2015; Vogel et al., 2017). This often exceeds the microbial thermal optimum, resulting in the death or dormancy of thermosensitive taxa (Donhauser et al., 2020; Riah-Anglet et al., 2015) and changes in the community composition (Bérard et al., 2011; Hawkes and Keitt, 2015). Consequently, this can trigger a shift in metabolism to facilitate microbial thermal acclimation and increase respiration in the remaining thermotolerant species (Anjileli et al., 2021; Bardgett and Caruso, 2020), inadvertently increasing organic matter mineralisation, depleting soil C stocks, and reducing microbial C use efficiency (CUE – the ratio of C allocated to growth over total C uptake; Allison et al., 2010; Ghee et al., 2013; Li et al., 2019; Mganga et al., 2022). As microbial CUE strongly influences soil C sequestration and is sensitive to various biotic (e.g., competition between species) and abiotic (e.g., pH, temperature, nutrient and organic C availability) factors (Iven et al., 2023; Jones et al., 2019), there is an urgent need to identify which management strategies may enhance the resilience of semi-arid Mediterranean soils to extreme heat-stress events (> 40 °C) to prevent widespread soil degradation (Ferreira et al., 2022).

Previous research has examined the impact of extreme heat waves on certain soils located in Mediterranean (Bañeras et al., 2022; Bérard et al., 2011). However, most studies do not assess the effects of extreme heat on these soils with challenging CUE conditions, such as calcareous soils in arid or semi-arid regions. These soils, on occasions (Vertisols and Alfisols) are characterized by periods of high moisture and reduced oxygen content, which tend to reduce microbial CUE (Zheng et al., 2019). Additionally, their high calcium carbonate content limits



microbial nutrient availability such as available P, further potentially decreasing CUE. This shift has accelerated the adoption of organic agricultural practices, such as compost application, to promote more sustainable and resilient farming systems (Moreno-Pérez, 2023), supported by European policies (Rato-Nunes et al., 2017). Recent research by Sánchez-Rodríguez et al. (2024) identified a 23-24 % reduction in microbial CUE in typical

Mediterranean soils (Inceptisol, Alfisol, and Vertisol) when fertilised with diammonium phosphate or single superphosphate. However, the reaction of soil microbes and alterations in microbial CUE due to organic amendments under similar conditions is still not clear. The composted bioamendments utilised in Mediterranean agroecosystems could offer a sustainable alternative to mineral fertilisers (e.g., diammonium phosphate) (Rodríguez-Espinosa et al., 2023) where nutrient-rich waste products such as olive mill pomace (Lozano-García

and Parras-Alcántara, 2013), biosolids (sewage sludge; Roig et al., 2012), and solid urban residue (e.g., municipal food waste; Pascual et al., 1997) are recycled to enhance soil quality and nutrient availability (Kok et al., 2023). Under ambient conditions (ca. 20-30 °C), bioamendments may improve agricultural productivity by altering microbial community composition and activity (Kok et al., 2023), enhancing extracellular enzymatic production for nutrient utilisation and microbial acclimation (Conant et al., 2011), providing essential energy for soil

microorganisms such as organic C, N and P (Wang and Kuyakov, 2023), and influencing soil biogeochemistry (Mooshammer et al., 2017). They may also promote improvements in soil structure and reduce soil bulk density and thus enhance plant growth. However, little is known about their impact on microbial CUE in agricultural soils, especially those subjected to extreme heat-stress events, highlighting a critical knowledge gap in understanding the use and potential benefits of bioamendments in Mediterranean regions with challenging soil properties (low

organic matter content and reduced availability of P) and subjected to extreme heat-stress events.

This study investigated the effect of various bioamendments (composted olive mill pomace, composted biosolids and composted solid urban residue) and a conventional fertiliser (diammonium phosphate) on CUE and soil biogeochemistry in a Spanish calcareous Vertisol and a non-calcareous Inceptisol with different physical and chemical properties and limited soil P availability subjected to extreme heat-stress. We hypothesise that i)

bioamendments will increase the availability of P and other nutrients supporting a more resilient soil microbial community with enhanced resistance to extreme heat-stress events than soils receiving conventional fertiliser; ii) this will then increase CUE and subsequently soil C retention in soils receiving bioamendments than the conventional P fertiliser; and iii) the calcareous Vertisol will provide a more buffered environment for microbial metabolism at higher temperatures than the non-calcareous Inceptisol due to the higher pH and clay content. To

critically assess this, different incubation experiments under controlled conditions were undertaken. In all cases, both soils were incubated for 7 to 9 days at 20, 30, 40 and 50 °C. Two incubation experiments included the addition of bioamendments (composted wastes) or a conventional fertiliser to the soils. In the first one, microbial activity (soil $CO_2$ production after adding $^{14}$C-glucose) was monitored after the heat stress, while in the second one (with the same experimental design as the first one but without $^{14}$C application), alteration in chemical properties were

evaluated. A third experiment was developed to evaluate the potential buffering effect of each soil at high temperatures and microbial activity was monitored after the heat stress in the lack of bioamendments and chemical fertilizer.






## 2. Materials and methods

### 2.1. Site description and experimental design

Soil samples (0-20 cm depth) were collected from two agricultural regions in Spain with different soil types. The first, a calcareous Vertisol located 19 km outside of Santa Cruz (Córdoba, Spain; 37º47'03''N, 04º36'35''W; 291 m a.s.l.) sown with wheat (*Triticum aestivum* L.), which is grown in rotation with sunflower (*Helianthus annus* L.), canola (*Brassica napus* L.) and legumes such as chickpea (*Cicer arietinum* L.). The second, a non-calcareous Inceptisol located 2.1 km outside of Casas de don Pedro (Badajoz, Spain; 39°07'11''N, 05°18'51''W; 393 m a.s.l.) within an established olive (*Olea europaea* L.) grove. These soils were selected for their different soil properties (e.g., pH, clay and carbonate content; Table 1), and as representative examples of agricultural soils in central and southern Spain. Additionally, both soils experience similar climatic conditions, characterized by prolonged periods of high temperature stress (Fig. S1). At each location (see Fig. S2), a homogenised, representative soil sample was collected for both soil types, then processed for analysis. After collection, soil was air-dried at room temperature (ca. 25 °C) for one-week before sieving to < 1 cm to remove large roots and stones (to use in subsequent incubation experiments), with a subsample of each soil type ground to < 2 mm for initial biogeochemical analysis. These locations were chosen as being representative of a hot Mediterranean climate (Csa, according to the Köppen classification), where air temperature regularly exceeds 40 °C and climate change driven temperature extremes and water stress are predicted to be at their most intense (Zagaria et al., 2023).

**Table 1.** Physicochemical characteristics of the Vertisol and Inceptisol. Values represent mean ± SEM, $n$ = 3 analytical replicates. Data is expressed on a dry weight basis where applicable.

| Soil physicochemical properties | Vertisol | Inceptisol |
|---|---|---|
| $pH_{1:2.5}$ | 8.06 ± 0.07 | 6.15 ± 0.16 |
| Electrical conductivity$_{1:5}$ ($\mu$S cm$^{-1}$) | 186 ± 2 | 54 ± 2 |
| Sand (g kg$^{-1}$) | 220 ± 5 | 570 ± 2 |
| Clay (g kg$^{-1}$) | 380 ± 8 | 120 ± 1 |
| Silt (g kg$^{-1}$) | 400 ± 7 | 310 ± 7 |
| Oxidizable organic carbon (g C kg$^{-1}$) | 11.5 ± 1.97 | 17.5 ± 0.36 |
| Total nitrogen (g N kg$^{-1}$) | 1.45 ± 0.34 | 1.76 ± 0.03 |
| C:N ratio | 7.93 ± 1.01 | 9.94 ± 0.21 |
| Carbonate (g kg$^{-1}$) | 392 ± 6 | <0.1 |
| Available phosphorus (mg P kg$^{-1}$) | 11.8 ± 0.1 | 8.3 ± 0.8 |
| Available iron (mg Fe kg$^{-1}$) | 3.2 ± 0.09 | 33.6 ± 3.30 |
| Available copper (mg Cu kg$^{-1}$) | 1.4 ± 0.02 | 0.6 ± 0.07 |
| Available manganese (mg Mn kg$^{-1}$) | 2.2 ± 0.1 | 182 ± 17.0 |
| Available zinc (mg Zn kg$^{-1}$) | 0.25 ± 0.04 | 0.23 ± 0.04 |

To explore the effect of conventional fertiliser *vs*. bioamendments on soil nutrient cycling and microbial activity under extreme heat-stress, a subsample (100 g; $n$ = 4 per treatment) of each soil type was mixed with either diammonium phosphate, applied as the conventional control (a common inorganic fertiliser), or a bioamendment



of composted olive mill pomace, biosolids, or solid urban residue (also referred to as municipal solid waste) (Table 2). The pH of the calcareous soil is buffered around 8 due to the presence of calcium carbonate, which limits nutrient availability including P. All treatments were applied at a rate of 50 mg P kg$^{-1}$ to reflect a typical agronomic P loading rate utilised in this region. Consequently, the soils received varying quantities of each respective bioamendment due to its underlying P content (Table 2), with a control treatment receiving no additions. The

obtained mixtures (soil/treatments) were wetted 1-week before incubation to ca. 0.18 g g$^{-1}$ gravimetric moisture content.

Composted olive mill pomace was obtained from the Vadolivo commercial olive oil mill in Cazorla (Jaén, Spain; 37°57'57.0"N, 03°10'23.1"W), where olive residue from the oil extraction process is combined with foliage (i.e., olive leaves) and local animal manure (e.g., cattle or goat manure, depending on availability) to aid decomposition.

Composted biosolids were obtained from the La Golondrina wastewater treatment plant (Córdoba, Spain; 37°50' 47.7"N, 04°51'52.2"W), which processes 148,602 m$^3$ d$^{-1}$ of waste from the local municipality of Córdoba, serving a population of ca. 330,000. Biosolids are initially processed via de-sanding, degreasing and sedimentation of solids, then subjected to biological oxidation, prior to centrifugal dehydration. Finally, composted solid urban residue, derived from local food waste and organic municipal residues, was obtained from the SADECO municipal

waste treatment plant (Córdoba, Spain; 37°53'09.1"N, 04°47'17.6"W). All materials were collected approximately 4-months before the start of the experiment, air-dried (ca. 22 °C) for one week, then stored in polyethene bags until use.

## 2.2. Local meteorological conditions

Daily meteorological data were recorded over a 20-year period (2003-2023) close to each soil sampling location

(Fig. S1 and S2; Agri4Cast, 2024). Annual precipitation recorded over the 20-year period was 424 ± 30.5 mm yr$^{-1}$ and 398 ± 23.9 mm yr$^{-1}$ for Córdoba and Badajoz cities, respectively (Fig. S3). For the Vertisol, meteorological data was obtained from a station located ca. 5.5 km from the soil sampling location near to Santa Cruz, approximately 27.0 km outside of Córdoba (37º45'46"N, 04º33'22"W; 246 m a.s.l). Here, daily air temperature regularly exceeded 30 °C within a 20-year period for an average of 120.3 ± 2.8 days yr$^{-1}$, within this period 9.7 ±

1.63 days yr$^{-1}$ exceeded 40 °C across a range of 2-29 days. For the Inceptisol, meteorological data was obtained from a station located ca. 15.5 km from the soil sampling location outside of Badajoz (38º52'45"N; 06º58'14"W; 113 m a.s.l.). Here, daily air temperature exceeded 30 °C within the 20-year period for an average of 107.3 ± 2.8 days yr$^{-1}$, within this period 6.4 ± 1.6 days yr$^{-1}$ exceeded 40 °C over a range of 0-28 days. At both sites, it is expected that soil surface temperatures during summer will reach 50-60 °C or higher due to intense solar radiation

and low soil moisture (Khorchani et al., 2018; Melo-Aguilar et al., 2022).

## 2.3. Soil physicochemical characteristics

Baseline soil physiochemical characteristics for the Vertisol and Inceptisol are presented in Table 1. Briefly, pH and electrical conductivity (EC) were determined using standard electrodes following a 1:2.5 w/v and 1:5 w/v (soil:solution) extraction, respectively, with deionized (DI) H$_2$O. Soil texture was determined via the Robinson

pipette method (Robinson, 1922). Oxidizable organic C content was determined by rapid dichromate oxidation



**Table 2.** Chemical composition of diammonium phosphate and the composted bioamendments. Values represent the mean ± SEM of $n = 3$ analytical replicates except for pH and electrical conductivity, which uses $n = 2$ analytical replicates. Data is expressed on a dry weight basis where applicable.

| Chemical composition | Diammonium phosphate | Olive mill pomace | Biosolids | Solid urban residue |
|---|---|---|---|---|
| $pH_{1:25}$ | 7.44 | 8.81 | 6.70 | 8.05 |
| Electrical conductivity$_{1:25}$ (mS cm$^{-1}$) | 0.29 | 1.00 | 2.71 | 2.59 |
| Volatile solid content (g kg$^{-1}$) | - | 750 ± 17 | 448 ± 14 | 562 ± 15 |
| Oxidable organic carbon (g C kg$^{-1}$) | 0.8 ± 0.01 | 474 ± 11 | 456 ± 39 | 639 ± 19 |
| Total soluble nitrogen (g N kg$^{-1}$) | 195.3 ± 1.4 | 20.6 ± 10.2 | 28.8 ± 1.1 | 32.5 ± 1.7 |
| C:N ratio | 0.004 ± 0.00 | 23.1 ± 0.5 | 15.8 ± 1.3 | 19.9 ± 0.6 |
| Total phosphorus (g P kg$^{-1}$) | 200 ± 0.02 | 0.51 ± 0.01 | 7.4 ± 0.3 | 1.91 ± 0.04 |
| Total copper (mg Cu kg$^{-1}$) | 44.0 ± 0 | 32.0 ± 0 | 538 ± 1 | 211 ± 1 |
| Total zinc (mg Zn kg$^{-1}$) | 59.0 ± 1 | 59.0 ± 1 | 589 ± 1 | 470 ± 1 |
| Loading rate (g kg$^{-1}$) | 0.4 | 44.7 | 3.2 | 12.6 |


Sodium bicarbonate-extractable P (Olsen-P) was measured colorimetrically using the molybdate blue method after extraction with 0.5 M NaHCO$_3$ at a pH of 8.5 (Olsen et al., 1954) and measured according to Murphy and Riley (1962). The micronutrient availability of iron, copper, manganese, and zinc was determined using the DTPA (diethylenetriaminepentaacetic acid) method (Fe$_{DTPA}$, Cu$_{DTPA}$, Mn$_{DTPA}$ and Zn$_{DTPA}$, respectively; Lindsay and

Norvell, 1978).

### 2.4.    Bioamendment chemical composition

The chemical composition of the synthetic fertiliser and composted bioamendments (Table 2) was determined utilising the methods provided by Thompson et al. (2002). Briefly, pH and EC were determined following a 1:25 w/v extraction with DI H$_2$O. Total N was measured using the Kjeldahl method (1883). Oxidizable organic C

content was determined via Walkley and Black (1934) method described previously after drying raw material ($n$ = 3 analytical replicates per material) at 105 ºC for 24-h. The volatile solids content (i.e., used as a proxy for organic matter) of each bioamendment was measured via loss-on-ignition at 550 °C for 2 h. The concentration of macro- and micronutrients were determined by acid digestion (Thompson et al., 2002).

### 2.5.    Microbial activity and carbon use efficiency

In this study microbial C uptake is defined as the total labelled C remaining in the system, which has not been respired as $^{14}CO_2$ or incorporated in the microbial biomass (Glanville et all., 2016; Sánchez-Rodríguez et al., 2024). The duration of each monitoring period was unique to each experiment and dependent on the gradual stabilisation in $^{14}CO_2$ emissions from microbial utilisation of the slow C pool. This technique captures $^{14}CO_2$



emissions from both catabolic (i.e., the rapid mineralisation) and anabolic (i.e., secondary slow $^{14}CO_2$ release due
to cell turnover) processes.

### 2.5.1.    Experiment 1: Microbial activity during and after an extreme heat-stress event

Microbial activity during and after an extreme heat-stress event was assessed by measuring microbial $^{14}CO_2$
release and CUE, following the methods described in Jones et al. (2019, 2018) in soils receiving bioamendments,
inorganic fertiliser, or no additions (control).  Briefly, 2.5 g of soil ($n$ = 5 per treatment, soil type and temperature)
was placed in a sterile 50 ml polypropylene centrifuge tube and rewetted with 200 µl of DI $H_2O$ 1-week prior to
performing a transient (7-day long) heat stress event.

After 1-week of pre-incubation at 20 °C, soils were placed in an incubator (LMS Incubator; Polestar Cooling Ltd.,
Bognor Regis, UK) at either 20, 30, 40 or 50 °C to reflect heatwaves of different intensity. After two days, 250 µl
of $^{14}C$-labelled glucose (4.6 kBq ml$^{-1}$, 10 mM; American Radiolabelled Chemicals Inc., St Louis, USA) was
pipetted evenly onto the soil surface. A 6 ml polypropylene vial containing 1 ml of 1 M NaOH was then placed
above the soil surface to capture respired $^{14}CO_2$ and the tubes sealed. Samples remained in the incubators for a
further 5-days after radiolabelling prior to returning to 20 °C. Following radiolabelling, NaOH traps were removed
and replaced at regular intervals until observation of a more stable shift in $^{14}CO_2$ emission (over 27-days: 20 °C
and 30 °C samples; day 0.04, 0.1, 0.2, 0.8, 1.0, 3.8, 4.8, 5.0, 5.1, 6.9, 10.9, 12.8, 14.9 and 27.0; and 21-days: 40
°C and 50 °C samples; day 0.1, 0.2, 0.3, 1.0, 2.0, 2.3, 5.0, 5.4, 6.0, 7.0, 9.1, 12.1 and 21.0). 4 ml of OptiPhase
HiSafe 3 scintillation fluid (Revvity Health Sciences B.V, Groningen, the Netherlands) was added to the NaOH
traps after removal to determine $^{14}C$ activity by liquid scintillation counting, using a Hidex 600SLE liquid
scintillation counter (Hidex Oy, Turku, Finland). After removal of the final NaOH trap, the amount of $^{14}C$
remaining in the soil was determined by extracting the soil with 12.5 ml of ice-cold 1 M NaCl (200 rev min$^{-1}$, 30
mins), centrifuging the extracts (18,000 g, 10 mins), and recovering 1 ml of the supernatant. The amount of $^{14}C$
remaining in the soil was then determined via liquid scintillation counting, as described previously.

Notably, since there is no universally accepted protocol for soil incubation experiments exploring extreme heat-
stress in the presence or absence of bioamendments (Schroeder et al., 2021), the duration of the extreme heat-
stress events conducted in this study was designed to reflect the typical duration and intensity of heatwaves
experienced in the region where the soil samples were collected, which can last over a week with daily air
temperatures reaching up to 45.4 °C (see Fig. S1). A mechanistic approach was utilised instead in our experiment,
where the soils were maintained at high temperatures for one week to explore microbial responses under an
extreme, worst-case scenario.

### 2.5.2.    Experiment 2: Bioamendments effect on soil chemical characteristics after an extreme heat-stress
235          event

To determine changes in chemical characteristics in soil receiving bioamendments or inorganic fertiliser after an
extreme heat stress event, other experiment was built in parallel to Experiment 1, with 2.5 g of soil ($n$ = 3 per
treatment, soil type and temperature) placed in a 1.5 ml Eppendorf tube, wetted with 200 µl of DI $H_2O$, and pre-
incubated at 20 °C for 1 week. Then the samples were heated to 20, 30, 40 or 50 °C for 9-days. Then, soil pH and
EC were determined on fresh soil following a 1:2.5 w/v (soil:solution) DI $H_2O$ extraction using standard
electrodes. Gravimetric soil moisture content was determined by drying soil at 105 °C for 24 h. Soil ammonium



($NH_4^+$) and nitrate ($NO_3^-$) were determined colorimetrically from 0.5 M $K_2SO_4$ extracts (1:5 w/v) using the methods described in Mulvaney (1996) and Miranda et al. (2001), respectively. Total extractable N and organic C were analysed using the 0.5 M $K_2SO_4$ extracts on a Multi N/C 2100 S analyser (AnalytikJena, Jena, Germany). Available phosphorus (Olsen-P) was determined using the method of Olsen et al. (1954) as described previously.

### 2.5.3. Experiment 3: The legacy effect of extreme heat-stress events on microbial activity in unamended soils

To understand the legacy effect of prolonged heat-stress, microbial CUE was explored after an extreme heat-stress event in another experiment where soils receiving no amendments (bioamendments or inorganic fertiliser) following a similar methodology described above. Briefly, 2.5 g of soil *(n = 5 per soil type and temperature, control soil only, without fertilization)* was placed in a sterile 50 ml polypropylene centrifuge tube and rewetted with 200 µl of DI $H_2O$ before allowing to pre-incubate for 1-week. After the pre-incubation period, soils were then placed in an incubator at 20 °C, 30 °C, 40 °C or 50 °C. Following a week of heating, soils were returned to 20 °C before 250 µl of $^{14}$C-labelled glucose (4.6 kBq ml$^{-1}$, 10 mM; American Radiolabelled Chemicals Inc., St Louis, USA) was pipetted evenly onto the soil surface. NaOH traps were placed above the soil surface and changed on days 0.04, 0.13, 0.33, 1, 2, 3, 6, 8, 15 and 16, prior to determining via liquid scintillation counting. Then, soil was extracted with fridge-cold 1 M NaCl to determine the amount of $^{14}$C remaining in the soil and microbial CUE calculated as described previously. In both experiments (Experiment 1 and 3), $^{14}CO_2$ measurements were completed and the soils extracted to determine microbial CUE once the rate of $^{14}CO_2$ evolution had plateaued, which explains the different length in each case and for different temperatures.

### 2.6. Statistical analysis

Data were analysed in R Studio (version 4.2.1) with graphical images produced using the 'ggplot2' package (version 3.3.6; Wickham, 2016). Prior to analysis, all data was tested for normality using the Shapiro Wilks test (R core stats package) and homogeneity of variance using the Levene's test ('car' package, version 3.1.1). If assumptions were not met following log10-transformation, then data were analysed using a non-parametric test (e.g., Kruskal-Wallis's test) where appropriate.

Data that met the assumptions for parametric testing were then analysed as follows. Microbial CUE under heat-stress and changes in soil biogeochemistry were analysed using a three-way ANOVA, with soil type, temperature and treatment assigned as the fixed factors, and their interaction (soil type × temperature × treatment) explored (Experiment 1 and 2). Significant interactions were then analysed using a post-hoc pairwise comparison. A two-way ANOVA was used to identify significant differences microbial CUE following heat-stressing, with temperature and soil type used as the fixed factors and their interaction (soil type × temperature) assessed (Experiment 3). Significant interactions were assessed using a pairwise comparison with a Bonferroni adjustment applied. Significance level was set at $p < 0.05$ for all statistical tests. Values presented in the text represent mean ± SEM unless otherwise stated.



## 3. Results

### 3.1. Experiment 1: Microbial activity during and after an extreme heat-stress event

#### 3.1.1. $^{14}$C-glucose mineralisation

Soil microbial respiration increased during extreme heat-stress events (Fig. 1, Table 3), with a critical shift observed when soils were subjected to 50 °C for 1-week (soil type × treatment × temperature; $F_{(12, 152)} = 1.910$, $p$ = 0.037). In the first 24 h following $^{14}$C-glucose addition, initial respiration rates increased from 16-32 % at 20 °C, 22-36 % at 30 °C, 19-38 % at 40 °C, to 40-61 % at 50 °C when exceeding the microbial temperature optima

(Table 3). Microbial respiration rates were often higher in the non-calcareous Inceptisol, rather than the calcareous Vertisol, and greater respiration ($p < 0.001$) was observed for each treatment at 50 °C than 20 °C (Table S1). While heat-stress was the primary driver of microbial respiration, the treatment (i.e., bioamendments *vs.* conventional fertiliser) had a limited significant effect at lower temperatures (e.g., 20 and 30 °C; Table S2), with no observable difference ($p > 0.05$) between treatments observed at 20 °C. In the Vertisol, significantly ($p < 0.001$)

greater initial microbial respiration rates were observed in the solid urban residue treatment than the control at 30, 40 and 50 °C, and in the biosolids treatment than the control at 40 °C (Table 3, 24 h after $^{14}$C-labelled glucose application). In the Inceptisol, the reverse trend occurred, with higher initial respiration rates in the control ($p < 0.05$) than in the diammonium phosphate and olive mill pomace treatments at 40 °C and all treatments at 50 °C ($p < 0.05$; Table 3, 24 h after $^{14}$C-labelled glucose application).

#### 3.1.2. Microbial carbon use efficiency after an extreme heat-stress event

Figure 1 shows the cumulative $^{14}CO_2$ losses as a function of soil (Vertisol and Inceptisol) temperature and treatment. The differences that were evident at 24 h or 5 days between treatments in each soil type × temperature combination after the application of $^{14}$C-labelled glucose were maintained until the end of the monitoring period. Significantly ($p < 0.05$) reduced microbial CUE was found in soils subjected to 50 °C, comparing to those at 20

°C, regardless of the soil type or treatment, except for the olive mill pomace that did not show a significant difference in microbial CUE at 20 and 50°C (Fig. 2; Table S3). Although a non-significant interaction between soil type × treatment × temperature was observed ($F_{(12, 152)} = 1.113$, $p = 0.353$), a clear shift in microbial CUE was identified, decreasing from 0.47-0.65 at 20 °C to 0.27-0.45 at 50 °C (Table S4). In general, microbial CUE was significantly higher in the Vertisol than in the Inceptisol ($p < 0.05$) at all temperatures and with all applied

treatments (except for olive mill pomace and solid urban residue, where no significant differences were recorded between the two soils at 50 °C). The addition of the bioamendments to both soils did not significantly affect microbial CUE at 20 and 30 °C comparing to the control in both soils, expect for the solid urban residue in the Vertisol, where it considerably decreased microbial CUE at 30°C when comparing to the control (Fig. 2 and Table S5). Otherwise, the differences in microbial CUE between applied treatments were more obvious with the increase

of the temperature. In the Inceptisol at 40 °C, microbial CUE was significantly higher at 40 °C in the samples treated with diammonium phosphate and olive mill pomace, and at 50 °C when olive mill pomace was added, in comparison with the control (Fig. 2).





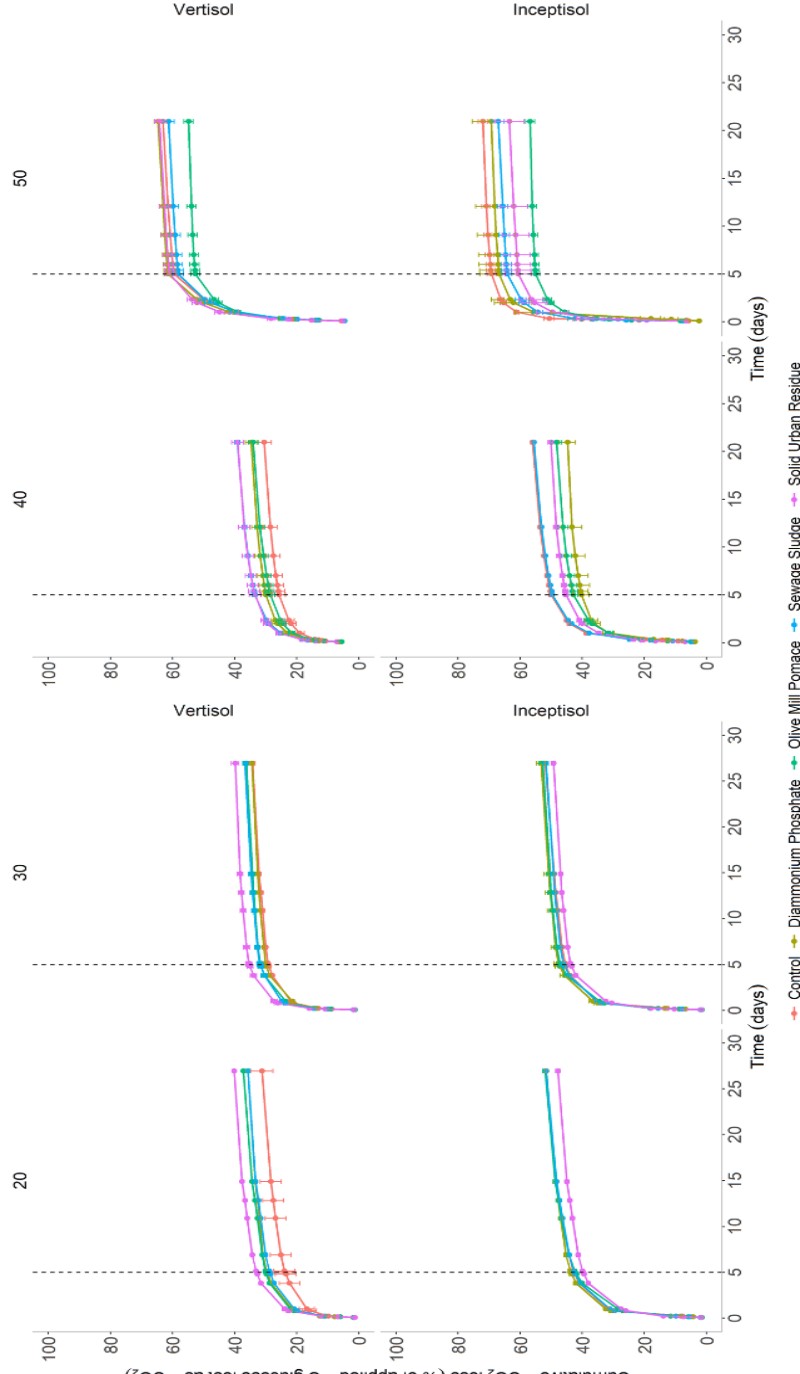

**Figure 1:** Experiment 1. Cumulative $^{14}CO_2$ loss (expressed as a % of the total applied $^{14}$C-labelled glucose lost as $^{14}CO_2$) in soils receiving either no addition (control), diammonium phosphate, or composted olive mill pomace, biosolids or solid urban residue subjected to heat-stress. Legend applies to both panels. Data represents mean ± SEM (n = 5 per treatment, soil type and temperature). Dashed line indicates end of heating on day 5 (7 days since the heat stress started) and sample return to 20 °C.



**Table 3.** Experiment 1. Total $^{14}CO_2$ respired (expressed as a % of the total $^{14}$C-glucose added) in the first 24-h after $^{14}$C-labelled glucose was applied to the soils receiving either no fertilization (control), diammonium

phosphate, or composted olive mill pomace, biosolids or solid urban residue. Data represent mean ± SEM ($n = 5$ per treatment, soil type and temperature).

| Treatment | 20 °C | | 30 °C | | 40 °C | | 50 °C | |
|---|---|---|---|---|---|---|---|---|
| | Vertisol | Inceptisol | Vertisol | Inceptisol | Vertisol | Inceptisol | Vertisol | Inceptisol |
| Control | 16.8 ± 2.43 | 31.6 ± 0.36 | 22.0 ± 0.37 | 35.6 ± 0.13 | 19.1 ± 1.55 | 38.7 ± 0.50 | 39.5 ± 0.74 | 61.3 ± 0.85 |
| Diammonium phosphate | 21.5 ± 0.27 | 32.5 ± 0.55 | 21.5 ± 0.73 | 36.6 ± 1.20 | 23.0 ± 1.25 | 31.5 ± 1.80 * | 41.8 ± 1.46 | 55.8 ± 5.83 * |
| Olive mill pomace | 22.2 ± 0.17 | 28.7 ± 0.32 | 23.7 ± 0.48 | 34.5 ± 0.38 | 21.6 ± 1.11 | 31.4 ± 1.20 * | 39.0 ± 0.98 | 45.7 ± 1.31 * |
| Biosolids | 20.8 ± 0.32 | 31.2 ± 0.42 | 24.8 ± 0.61 | 35.4 ± 0.45 | 25.2 ± 0.54 * | 37.8 ± 0.37 | 39.8 ± 1.59 | 54.4 ± 1.61 * |
| Solid urban residue | 24.0 ± 0.65 | 27.6 ± 0.37 | 27.3 ± 1.02 * | 32.4 ± 0.30 | 25.8 ± 0.92 * | 34.8 ± 0.82 | 44.9 ± 1.52 * | 49.6  ± 5.12 * |

*Significant difference detected ($p < 0.05$) of the obtained results between the composted bioamendments or diammonium phosphate and the control as determined by the Pairwise Comparison (PWC) test.*




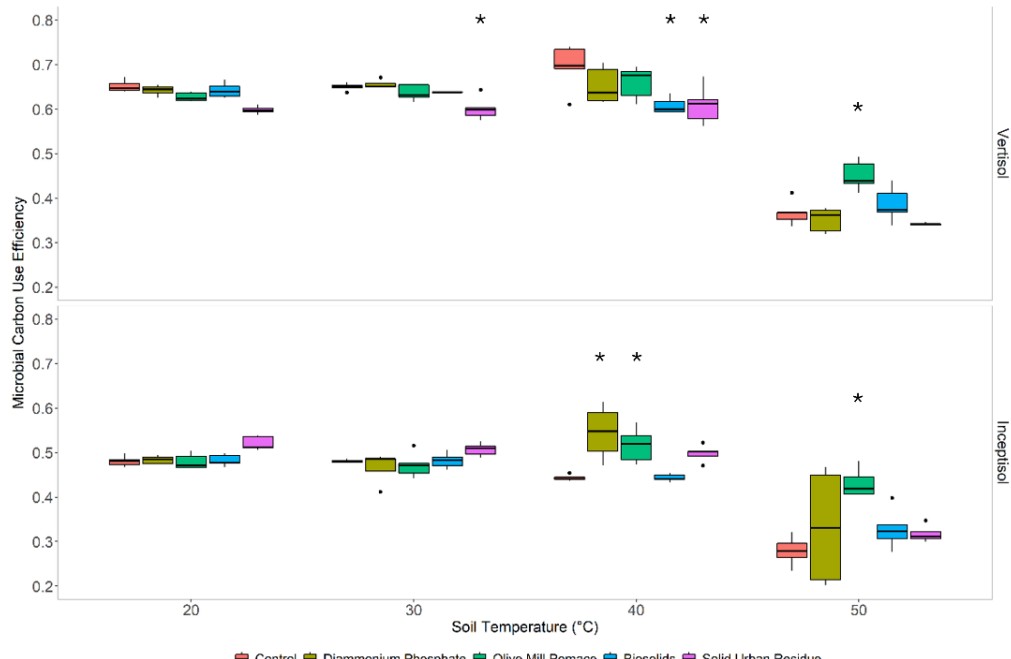

**Figure 2:** Experiment 1. Microbial carbon use efficiency (CUE) determined in soils receiving either no addition (control), diammonium phosphate, or composted olive mill pomace, biosolids or solid urban residue subjected to heat-stress ($n$ = 5 per treatment, soil type and temperature). Legend applies to both panels. Boxplots display the median and interquartile range, with whiskers showing minimum and maximum values in the data, and dots indicating potential outliers. Asterisks (*) indicate significance of differences ($p < 0.05$) of the obtained results between the composted bioamendments and the control as determined by the Pairwise Comparison (PWC) test.



### 3.2. Experiment 2: Soil chemical composition after an extreme heat-stress event

#### 3.2.1.    pH and electrical conductivity

Despite a significant three-way interaction between soil type, treatment and temperature ($F_{(12, 78)}$ = 3.450, $p <$ 0.001), soil pH was unaffected in the Vertisol (Fig. S4, top panel), remaining at ca. pH 8-8.5 after the heat-stress, whereas pH in the Inceptisol was strongly affected by both temperature and treatment ($p < 0.001$). In the Inceptisol (Fig. S4, bottom panel), olive mill pomace significantly ($p < 0.001$) increased soil pH from acidic to neutral at 20 °C, 30 °C and 40 °C compared to other amendments. At 50 °C, no significant difference was observed between treatments, with soil pH remaining neutral (ca. pH 6.5-7.5). Moreover, no significant three-way interaction was observed for soil electrical conductivity ($F_{(12, 79)}$ = 1.578, $p = 0.115$), however, increasing the temperature from 40 °C to 50 °C in the Vertisol increased ($p < 0.001$) soil electrical conductivity in the soils treated with diammonium phosphate and olive mill pomace.

#### 3.2.2.    Soil carbon and nitrogen cycling

Soil extractable total dissolved organic carbon (TOC) had a variable response to extreme heat-stress, with no significant three-way interaction detected between soil type, treatment, and temperature ($F_{(12,80)}$ = 0.411, $p = 0.955$; Fig. 3). Although extreme heat-stress did not consistently affect TOC in the control, olive mill pomace and solid urban residue significantly increased TOC in both soils for all temperatures. Extreme heat-stress also had an inconsistent effect on soil total extractable nitrogen (TN), with no significant three-way interaction detected (soil type × treatment × temperature; $F_{(12,80)}$ = 1.063, $p = 0.402$). The highest dissolved TN was often observed in the diammonium phosphate treated soil, regardless of the temperature (Fig. 3), with concentrations significantly increasing from 97.1 ± 8.19 *vs.* 94.1 ± 9.22 mg N kg$^{-1}$ DW at 20 °C to 141 ± 7.86 *vs.* 142 ± 5.54 mg N kg$^{-1}$ DW at 50 °C in the Vertisol and Inceptisol, respectively ($p < 0.001$).

Soil $NH_4^+$ concentration generally increased under extreme heat-stress ($F_{(12, 80)}$ = 4.531, $p < 0.001$; soil type × treatment × temperature; Fig. S5), peaking at 50 °C in the diammonium phosphate treatment in the Vertisol and Inceptisol (86.6 ± 9.59 *vs.* 107 ± 3.34 mg $NH_4^+$-N kg$^{-1}$ DW, respectively). A key shift in soil $NH_4^+$ concentration was observed in the Vertisol and Inceptisol with a significant increase observed between 20 °C and 50 °C for all treatments ($p < 0.05$), except olive mill pomace (Fig. S4). In comparison, soil $NO_3^-$ concentration in the Inceptisol was unaffected by extreme heat-stress events, regardless of the temperature or treatment ($p > 0.05$; Fig. S5). In the Vertisol, soil $NO_3^-$ concentration in the diammonium phosphate treatment significantly decreased between 20 °C (79.4 ± 7.73 mg $NO_3^-$-N kg$^{-1}$ DW) and 50 °C (25.5 ± 3.20 mg $NO_3^-$-N kg$^{-1}$ DW) ($p < 0.05$).





365

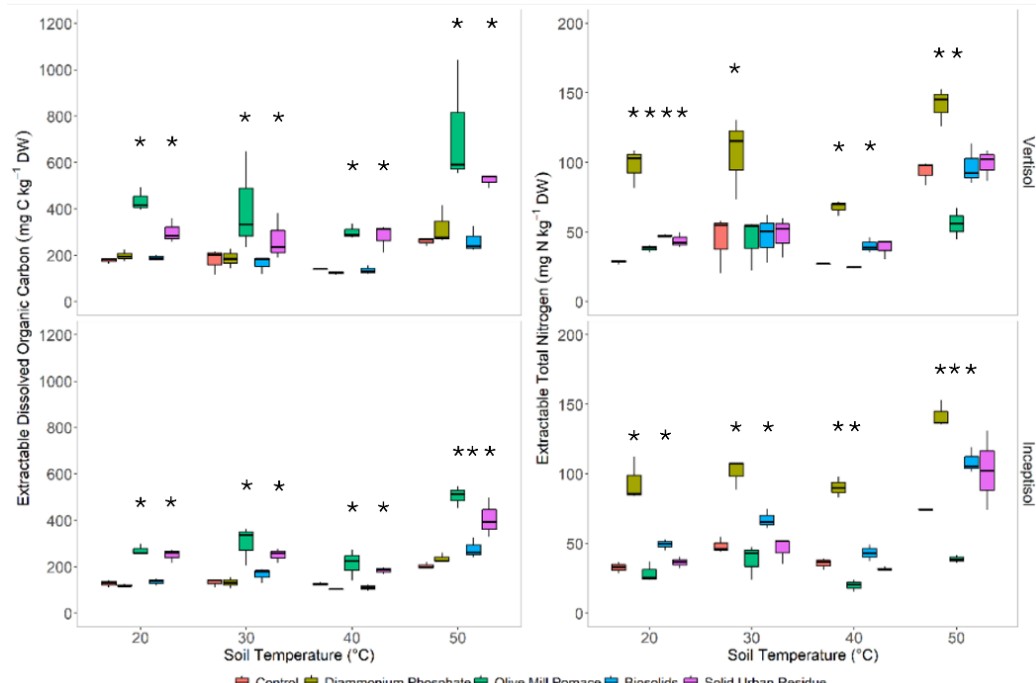

**Figure 3:** Experiment 2. Soil extractable dissolved organic carbon and nitrogen determined 9-days after heat-stressing soils receiving either no addition (control), diammonium phosphate, or composted olive mill pomace, biosolids or solid urban residue to 20, 30, 40 or 50 °C (n = 3 per treatment, soil type and temperature). Legend

370    applies to both panels. Boxplots display the median and interquartile range, with whiskers showing minimum and maximum values in the data. Asterisks (*) indicate significance of differences (p < 0.05) of the obtained results between the composted bioamendments and the control as determined by the Pairwise Comparison (PWC) test.



### 3.2.3. Available phosphorus

Olsen-P was unaffected or slightly affected by extreme heat-stress events, with a non-significant three-way interaction between soil type × treatment × temperature observed after 9-days of heat-stress ($F_{(4,100)} = 0.118$, $p = 0.118$). A significant two-way interaction was obtained between treatment and temperature in both soils ($F_{(4,50)} = 9.00$, $p < 0.001$- Inceptisol; $F_{(4,50)} = 4.79$, $p = 0.002$- Vertisol). Olsen-P significantly ($p < 0.05$) decreased in the diammonium phosphate treatment in both the Vertisol and Inceptisol (Fig. 4) with increasing temperature.

This decline was greatest in the Inceptisol, where soil P concentrations in the diammonium phosphate treatment decreased by 43 % from 34.1 ± 3.97 mg P kg$^{-1}$ DW at 20 °C to 19.4 ± 1.20 mg P kg$^{-1}$ DW at 50 °C. However, biosolids significantly increased Olsen-P in the Inceptisol in comparison with the control across all temperatures (Fig. 4).

### 3.3. Experiment 3: Microbial activity and carbon use efficiency after an extreme heat-stress event

Both soil types showed a strong resistance to high temperature (up to 40 °C), however a considerable legacy effect was observed on both soil types at 50°C, with unamended soils receiving $^{14}$C-glucose after heating for 1-week then returning to 20 °C displaying poor resilience to extreme heat-stress (50 °C). After 1-week, greater microbial respiration occurred in soils subjected to 50 °C than those heated to 20, 30 or 40 °C ($F_{(3, 30)} = 4.228$, $p = 0.013$; Fig. S6). Microbial respiration, measured in the first 24-h after heat-stress events had occurred, was significantly greater ($p < 0.001$) in the Vertisol subjected to 50 °C (26.5 ± 0.82 %) than 20 °C (19.0 ± 0.20 %), 30 °C (19.4 ± 0.36 %) or 40 °C (17.9 ± 1.47 %). Although initial respiration rates were slightly lower in the Vertisol, there was a lag-phase in the Inceptisol, where significant differences ($p < 0.001$) between each level of heat-stress only occurred after 48-h, reaching a maximum of 47.7 ± 0.90 % at 50 °C *vs.* 35.8 ± 0.66 % at 20 °C, 40.7 ± 0.85 % at 30 °C, and 36.3 ± 2.13 % at 40 °C. Although no significant two-way interaction between soil and temperature on microbial CUE was identified following exposure to heat-stress ($F_{(3, 30)} = 1.081$, $p = 0.372$; Fig. 5), temperature had a strong significant effect ($F_{(3, 30)} = 108.3$, $p < 0.001$), with a pairwise comparison detecting significantly lower microbial CUE values ($p < 0.001$) between soils (Vertisol and Inceptisol) subjected to 50 °C *vs.* 20 °C, 30 °C, and 40 °C.

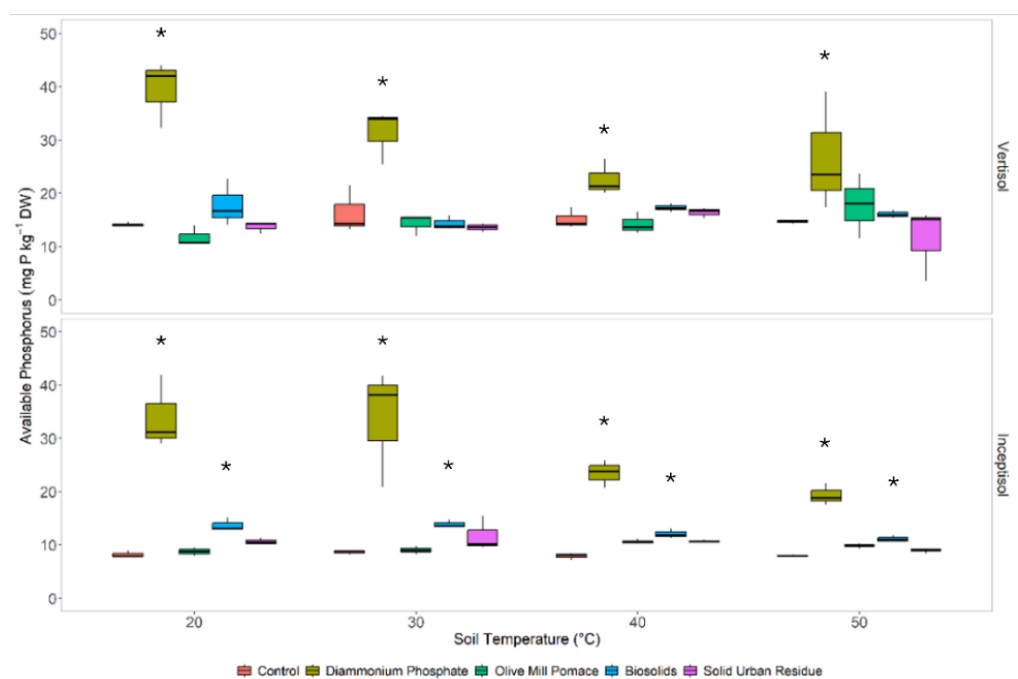

**Figure 4:** Experiment 2. Soil available P (Olsen-P) determined 9-days after heat-stressing soils receiving either no addition (control), diammonium phosphate, or composted olive mill pomace, biosolids or solid urban residue to 20, 30, 40 and 50 °C ($n$ = 3 per treatment, soil type and temperature). Legend applies to both panels. Boxplots display the median and interquartile range, with whiskers showing minimum and maximum values in the data. Asterisks (*) indicate significance of differences ($p < 0.05$) of the obtained results between the composted bioamendments and the control as determined by the Pairwise Comparison (PWC) test.



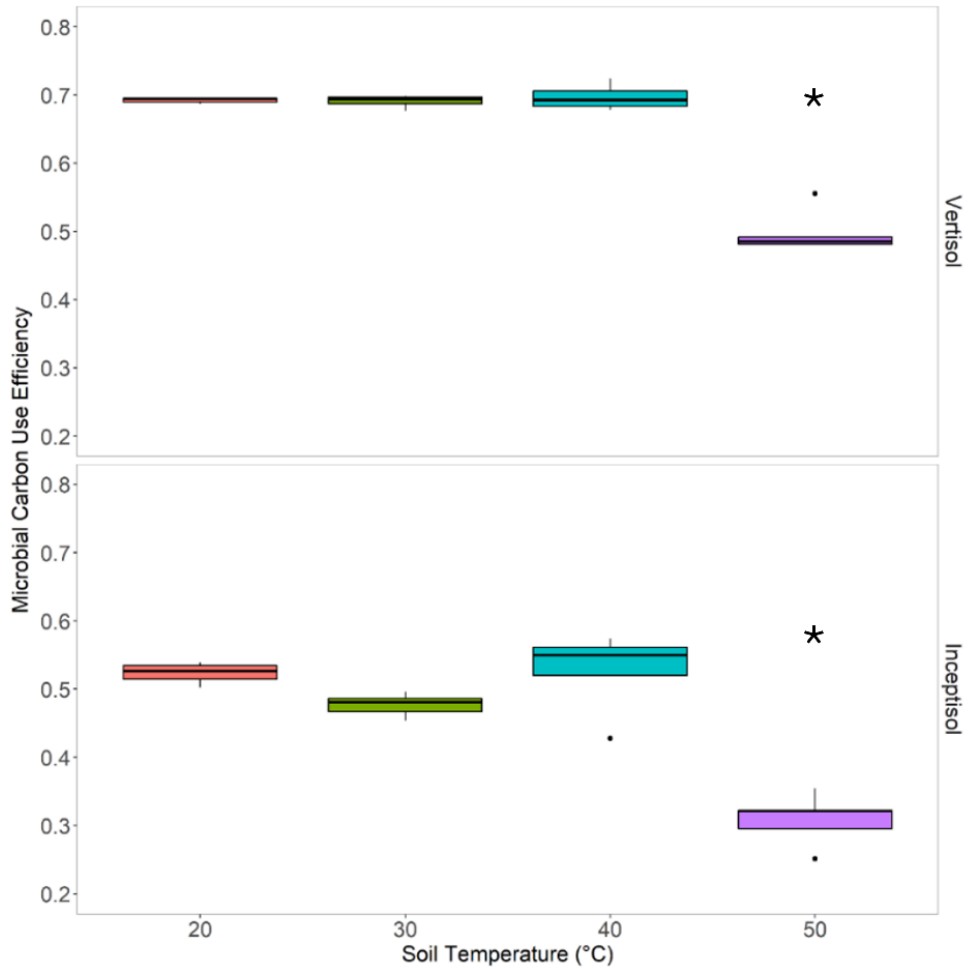

**Figure 5:** Experiment 3. Microbial carbon use efficiency (CUE) after the incubation period following one week
of heat-stress at 20, 30, 40 or 50 °C ($n$ = 5 per soil type and temperature). Boxplots display the median and
interquartile range, with whiskers showing minimum and maximum values in the data, and dots indicating
potential outliers. Asterisks (*) indicate significance of differences ($p < 0.05$) of the obtained results between the
composted bioamendments and the control as determined by the Pairwise Comparison (PWC) test.



## 4. Discussion

### 4.1. Impact of extreme heat-stress on soil nutrient cycling, microbial activity and CUE

Previous studies have shown that soil microbial respiration increases exponentially with temperature (Karhu et al., 2014; Qiao et al., 2019). However, many of these studies rarely exceed > 35 °C and are typically focussed on
longer-term warming in temperate (Frey et al., 2013; Ghee et al., 2013; Kok et al., 2023; Li et al., 2019; Riah-Anglet et al., 2015; Steinweg et al., 2008; Tucker et al., 2013), alpine (Donhauser et al., 2020; Wu et al., 2015), and sub-artic (Tájmel et al., 2024) environments. Studies with soils from Mediterranean areas, are often overlooked (Anjileli et al., 2021; Bérard et al., 2011; Hamdi et al., 2011). For that, our study on soils from Spain with limiting availability of P and treated with bioamendments are of interest to understand microbial CUE in
these semi-arid areas. Moreover, there are not many studies that evaluate the role of soil type and different amendments in buffering the negative effect of extreme heat events on CUE, especially, when soils are exposed to extreme heat stress where air temperatures exceed 40 °C (Fig. S1).

The lower $CO_2$ released and the higher microbial CUE obtained in the Vertisol compared to that in the Inceptisol at different temperatures could be explained due to the inherent properties of the Vertisols, which are characterized
by a buffered pH around 8.0. This basic pH favours microbial community richness in comparison with neutral or acidic soils, which facilitate a more efficient use of C (Zhu et al., 2024). In addition, the Vertisol used in this study had a higher nutrient availability as seemed in the slightly higher P availability in soil (Olsen P; 12 *vs*. 8 mg kg[-1]) and the lower C:N ratio (8 *vs*. 10 mg kg[-1]; Table 1) than the Inceptisol, which should have contributed to increase microbial CUE (Manzoni et al., 2012). Additionally, the higher clay content in the Vertisol (i) limits the
accessibility of microbes to the substrate which enhance CUE and slows down C use and $^{14}CO_2$ production (He et al., 2024) in comparison with the edaphic conditions for soil microorganisms in the Inceptisol, and (ii) should have played a considerable role in protecting soil microorganisms under heat-stress.

According to previous research that are partially in line with our findings (Qiao et al., 2019; Ren et al., 2024), temperature plays a critical role in soil microbial activity. In this respect, significant differences in microbial CUE
were observed in the Vertisol with solid urban residue (reduced CUE) at 30 and 40 °C and biosolids at 40 °C (reduced CUE), in the Inceptisol with diammonium phosphate and olive mill pomace at 40 °C (increased CUE) and with olive mill pomace at 50 °C (maintained the highest values of CUE) in both soils (Fig. 2). Moreover, in the lack of treatments (DAP or bioamendments; Fig. 5), significant differences were only found at 50 °C (the lowest microbial CUE in comparison with the rest of the temperatures). This could indicate that the soils used in
this study are adapted to high temperatures (up to 40 °C), probably due to their previous history of heatwaves and heat-stress. Additionally, thermal acclimation incurs a greater metabolic cost (Iven et al., 2023), which is in line with our results on microbial CUE in the control soils that declined at a rate of $0.010 \pm 0.001$ °C[-1] and $0.007 \pm 0.001$ °C[-1] in the Vertisol and Inceptisol, respectively, falling within a similar range reported by Tucker et al. (2013) (from -0.0011 to -0.017 °C[-1]) and Steinweg et al. (2008) (ca. -0.009 °C[-1]) in a high-elevation sagebrush
and lowland grassland ecosystem type, respectively. The effect of the different treatments on microbial CUE depended on the soil and treatment and is explained with microbial respiration.



The rapid increase in microbial respiration observed in the first 24-h following [14]C-glucose addition, both during and immediately after an extreme heat-stress event at 50 °C in the two soils, indicates a significant shift in the microbial allocation of resources from growth and maintenance to survival-related metabolism to facilitate acclimation (Kok et al., 2023; Schimel et al., 2007). As thermal acclimation can encompass individual physiological adaptations (e.g., production of heat shock proteins and osmoprotectants, shifts in lipid membrane

composition) (Hall et al., 2010; Welsh, 2000) and community level changes (e.g., death of taxa, competition between species) across a variety of temporal scales (e.g., days to weeks) (Bardgett and Caruso, 2020; Bradford, 2013; Hawkes and Keitt, 2015), it should be noted that this study likely only captured the initial shift in microbial community composition and function, not full acclimation, due to [14]C-glucose substrate limitation. Additionally, the effect of the temperature on microbial CUE was as a function of the soil and treatment. Composted solid urban

residue at 30, 40 and 50 °C and composted biosolids at 40 °C increased [14]CO$_2$ production in the Vertisol having negative consequences on some occasions in CUE (Fig. 2), probably because of their lower C:N ratio, which should have fuelled soil microorganisms' activity (Table 2). Nevertheless, the opposite effect was observed in [14]CO$_2$ production in the Inceptisol at 40 °C with DAP or composted olive mill pomace, and with DAP and the three bioamendments at 50 °C (Table 3 and Fig. 1), which indicates the complexity of predicting microbial CUE

in these soils when bioamendments are added and heat-stress occurs. The obtained results confirmed those of Ren et al. (2024), indicating that soil microbial respiration can be adjusted according to the ambient conditions, including soil properties, temperature, and treatments. In our study, the immobilisation of soil enzymes by clay particles in the Vertisol (higher clay content) may have reduced denaturation of soil enzymes (including phosphatases) at high temperatures (Burns et al., 2013; Gianfreda and Ruggiero, 2006). This allows immobilised

enzymes to retain their catalytic activity above the threshold for free enzyme denaturation (i.e., > 60 °C) (Skujiņš, 1976; Sarkar et al., 1989). This could explain part of the differences observed in microbial CUE (reduced with certain bioamendments; Fig. 2) observed in the two soils used in this study. In addition, although the higher soil carbonate concentration buffered changes in soil pH in the Vertisol (Fig. S4), it should have had a limited significant effect on soil respiration and microbial CUE, with these differences instead primarily driven by

temperature and bioamendments addition, allowing us to partially reject our third hypothesis under the conditions of the study (the calcareous Vertisol will provide a more buffered environment for microbial metabolism at higher temperatures than the non-calcareous Inceptisol due to the higher pH and clay content; higher microbial CUE was obtained in the Vertisol at all temperatures except al 50 °C).

The application of the bioamendments modulated the effect of heat stress on both soils and on the availability and

the turnover of nutrients. At 50 °C, soil respiration and microbial CUE values were the closest to those observed at lower temperature when olive mill pomace was applied (20, 30, and 40 °C). This suggests that olive mill pomace improved both soils microbial resistance to extreme heat scenarios. The high amount of volatile solids (a proxy for organic matter) added to the soil by this bioamendments due to its low P content, which required a larger application rate to meet the target of 50 mg P kg$^{-1}$, resulted by introducing more organic C into the system. This

was highly reflected in the consistently elevated levels of extractable dissolved organic C across all temperatures (Fig. 3), which may have supported microbial metabolic needs under thermal stress. Consequently, this could have promoted the production of extracellular enzymes involved in organic matter decomposition (Conant et al., 2011; Mooshammer et al., 2017), enabling microbial communities to maintain function despite exceeding their thermal



optimum (Kok et al., 2023). In addition to C, the olive mill pomace treatment also resulted in relatively stable levels of extractable TN in both soils, while the diammonium phosphate treatment showed a marked increase in TN, particularly at 50 °C (Fig. 3). According to previous research (Dai et al., 2020), the application of fertilisers, especially diammonium phosphate, leads to a greater accumulation of $NH_4^+$ (Fig. S5) and dissolved TN. This likely reflects either increased mineralization of organic N or decreased microbial assimilation under stress. Meanwhile, $NO_3^-$ levels remained relatively stable in the Inceptisol, but decreased in the Vertisol under diammonium phosphate application at 50 °C, possibly due to denitrification or inhibited nitrification processes. These changes in N pools may partially explain the observed differences in microbial CUE, while $NH_4^+$ accumulation could aid microbial survival under heat stress, it may also lead to metabolic imbalance if C:N ratios become suboptimal. Similarly, $NO_3^-$ depletion may indicate altered N transformation pathways and reduced N use efficiency, ultimately lowering microbial CUE. These findings support our first and second hypotheses regarding the benefits of bioamendments (particularly composted olive mill pomace), for enhancing nutrient availability and microbial resilience under heat stress. However, the same positive effects were not observed for composted biosolids and solid urban residue in the Vertisol, where CUE significantly declined at 40 °C. This highlights the importance of the interaction between bioamendment type, temperature, and soil properties, especially the differing organic C content among treatments, which likely modulates microbial responses to extreme temperatures.

Besides the effectiveness of bioamendments in increasing soil resistance to extreme heat conditions, it is important to mention its role in maintaining the stable availability of some primordial nutrients in the soil even with the most extreme heat conditions. The added bioamendments kept the levels of available P relatively high at 40 and 50 °C in comparison with the control, observing significant increases with the composted biosolids for all temperatures in the Inceptisol (Fig. 4), probably related to the presence of carbonates and the buffered pH in the Vertisol (reducing the availability of P and other nutrients). Composted biosolids and composted solid urban residue are usually rich in carbohydrates and amino-sugars (Monda et al., 2017; Ferraz-Almeida et al., 2015), resulting in a relative fast mineralization rate. In addition, the presence of polyphenols in the composted olive mill pomace could have reduced its mineralization rate. Furthermore, it is also observed a decline in soil available P with increasing temperatures in both soils when diammonium phosphate was applied (Fig. 4), with this effect further confounded by abiotic processes (e.g., sorption of P to iron oxides and clay, precipitation, immobilisation) exacerbated by extreme heat-stress events (Hou et al., 2018; Tian et al., 2023). This should also have a role on reducing the availability of P released from the bioamendments but probably to a lower extent than when the chemical fertilizer was applied. It is well known that the bioamendments are gradually mineralized, providing nutrients (P, among others) while they are decomposed by soil microorganisms, which reduces the effect observed for diammonium phosphate and soil available P with increasing temperatures.

### 4.2. Wider implications and priorities for future research

This study highlights the vulnerability of soils located in semi-arid regions to extreme heat-stress events, with our results emphasizing the importance of carefully managing soil fertility to mitigate the cascading impacts of climate change on soil ecosystem service delivery (Qu et al., 2024). Across Central Europe, soil heat extremes are raising their frequency and intensity, as observed in the increase in the surface air temperature extremes by ca. 0.7 °C per decade (García-García et al., 2023), leading to significant environmental and socioeconomic implications (García-



León et al., 2021). As the severity of heatwaves is projected to worsen under current climate change scenarios (Seneviratne et al., 2021), it is vital that future studies utilise a holistic approach, encompassing both microbial and biogeochemical assays, to identify optimal agronomic practices (e.g., application rates, timings) for bioamendments to buffer soils to extreme heat-stress events and prevent loss of soil nutrients and organic matter (Bernard et al., 2022; Ghee et al., 2013). Additionally, we have demonstrated here the complexity of predicting the potential positive effects of applied bioamendments across various soil types. Further research is needed at various spatiotemporal scales to fully capture the impact of recurring heatwaves on soil biogeochemical cycling, as this is currently unknown. This will further improve the accuracy and development of region-specific soil C and global-scale Earth system models (Moinet et al., 2021; Qu et al., 2024; Ren et al., 2024).

However, it is important to note that while this study provides a valuable insight into the future response of Mediterranean soils to climate change, it may not accurately represent *in-situ* responses where soils are subjected to other confounding biotic (e.g., presence or absence of plants) or abiotic stressors, (e.g., drought, UV stress, and diurnal temperature fluctuation). In particular, while our experiment-maintained soil gravimetric moisture content at 0.1-0.2 g g$^{-1}$ (see Supplementary Materials for more information), removing any potential effect of moisture stress, we did not account for daily air temperature fluctuations that occur during an extreme heat-stress event, where air temperature in these regions can vary by ± 20 °C (Fig. S1). Consequently, soils then missed crucial periods of 'rest' from extreme heat-stress that may have otherwise enabled greater thermal acclimation of the microbial community. Therefore, we recommend that our results are considered on a purely mechanistic basis. As such, future studies would benefit from deploying isotopic tracing techniques (e.g., $^{15}$N, $^{18}$O, or $^{14}$C) and biological assays (e.g., enzymatic activity, 16S rRNA sequencing, transcriptomics) where soils are subjected to natural or artificial diurnal temperature fluxes to explore these confounding factors in greater detail.

## 5. Conclusions

Extreme heat-stress events can cause significant dysregulation of soil nutrient cycling and microbial activity in soils located in semi-arid regions, such as the Vertisol and Inceptisol evaluated here, with a critical thermotolerance threshold identified at 50 °C. Despite their climatic history, both soils demonstrated high resistance and resilience to high temperatures (30 to 40 °C), but the Vertisol exhibited even greater resistance (higher CUE at different temperatures). However, both soils were less effective under extreme heat stress events (> 40 °C) regardless of the treatment or underlying soil biogeochemistry, with a prolonged legacy effect on microbial respiration and microbial CUE observed. Bioamendments supplying a higher organic matter content to meet P supply, such as composted olive mill pomace, may support greater thermal acclimation of the microbial community at 50 °C compared to a conventional fertiliser, as found in our study. Moreover, the effect of the different bioamendments was a function not only on the soil but also on the temperature and bioamendment combination, which indicates the complexity of the processes involved in microbial CUE. Surprisingly, only the composted biosolids had a positive impact on soil P bioavailability in the Inceptisol under high to extremely high heat-stress under the conditions of the study. Therefore, longer term studies with recurring stresses (e.g., drought, heat-stress, flood) and repeated substrate inputs (e.g., $^{14}$C-glucose or plant litter) coupled with microbial community composition and function measurements (e.g., metagenomic sequencing, PLFAs, enzymatic activity) are needed to capture the acclimation and recovery phase of these soils and any subsequent shifts in their baseline





ecosystem services. Our results will support the optimisation and use of composted bioamendments in high-risk Mediterranean agroecosystems in soil with low organic matter content, further enabling the development of region-specific nutrient management guidelines to mitigate climate change.



**Acknowledgements**

The authors would like to thank the research technicians at the University of Córdoba Soil Science unit (AGR-165), Ms. Esther Blanco Mengual and Ms. Ester Anta Domínguez, for their invaluable assistance and support conducting this experiment. Graphical abstract was produced using Biorender.com.

**Funding**

This study was funded by the Ministry of Science, Innovation and Universities (reference: PID2020-118503RB-C22 'FerPhOS') and the Spanish National Research Agency through the Severo Ochoa and María de Maeztu Program for Centers and Units of Excellence in R&D (reference: CEX2019-000968-M). Ms. Sana Boubehziz is supported by the Spanish Government with a Research Staff Training Fellowship (FPI-2020, reference: PRE2020-092577). Dr Emily Cooledge, Prof Dave Chadwick and Prof Davey Jones did not require external funding support.

**Author contributions**

Funding acquisition: A.R.S.R., V.B. Conceptualisation, experimental design: S.B., E.C.C. D.R.C., A.R.S.R., D.LJ. Sampling, formal data analysis: S.B., E.C.C. Writing - first draft: S.B., E.C.C. Writing - review and editing: S.B., E.C.C., A.R.S.R., V.B., D.R.C, D.LJ. Supervision: E.C.C., A.R.S.R., V.B., D.R.C., D.L.J.

**Declaration of interest**

The authors declare no conflict of interest.

**Ethical Statement**

No ethical approval required.



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
