# Peer review of "Do composted bioamendments enhance the resistance of Mediterranean agricultural soils and their microbial carbon use efficiency to extreme heat-stress events?"

_EGUsphere, 2025_

## Author Comment (AC1)

**General**

**The manuscript is about the effect of a high soil temperature for a certain period on microbial activity ($^{14}CO_2$ from glucose) and microbial carbon use efficiency ($^{14}C$ growth/ $^{14}C$ uptake).**

**Response:** Firstly, we would like to thank the reviewer for the constructive and detailed feedback that helped us improve our manuscript. Below, Reviewer#1 can find our point-by-point responses, with clarifications, justifications, and, where applicable, modifications in the manuscript.

**The hypothesis -I think- is that heat results in a lower CUE, and that balanced nutrient supply or organic amendments decrease the magnitude of the lowering of the CUE.**

**Response:** To clarify, our hypotheses were stated in the Introduction section of the manuscript. They are:

i) bioamendments will increase the availability of P and other nutrients supporting a more resilient soil microbial community with enhanced resistance to extreme heat-stress events than soils receiving mineral fertiliser;

ii) the enhancement of soil heat resistance will then increase or maintain CUE and subsequently soil C retention in soils receiving bioamendments than the mineral P fertiliser; and

iii) the calcareous Vertisol may exhibit greater thermal and chemical buffering capacity under extreme heat events, supporting microbial metabolism at elevated temperatures more effectively than the non-calcareous Inceptisol, due to its higher pH and clay content, which help retain moisture.

We would say that our two first hypothesis are in line with the hypothesis that Reviewer#1 mentioned here.

**If CUE is still high at 50 C, the authors assume that the soils has a high resistance to heat. Two soils are tested and three amendments.**

**Response:** We partially disagree here with Reviewer#1. CUE was reduced in both soils at 50 °C in comparison with 20, 30 and 50 °C as can be seen in Figure 2. It happened for all soils except soils treated with composted olive mill pomace, which had the highest microbial CUE

at this temperature (50 °C) in comparison with the rest of treatments (increase in soil resistance to heat).

**The type of microbes in the soils were not determined.**

**Response:** Exactly, in this study we focused on nutrient availability in soil, $^{14}CO_2$ emissions and microbial CUE as a function of the soil (Vertisol and Inceptisol) and treatment (control, Diamonium Phosphate-mineral fertiliser and three composted amendments).

**Specific comments**

**The introduction is not always logical, and reading the text still gives many small questions which is unnecessary.**

**Response:** Thanks for this comment. We have deeply modified the Introduction section according to Reviewer#1 comments, including a re-structuration to enhance clarity and logical flow. Several ambiguous phrases and words (e.g., "poor", "negative impact") were also modified. Please, see our responses to the rest of comments related to the Introduction section in the new version of the manuscript.

**I wonder if the method is correct. CO2 can be precipitated by Ca and Mg rich materials at certain CO2 concentrations. The vertisols but also the composts probably contain carbonates. Should this be tested with a dead soil?**

Response: Although we understand Reviewer#1 concern, this method is commonly used with soils containing carbonates to the effect of different evaluate edaphoclimatic conditions and fertilisation strategies on soil microbial CUE. In addition, Strom et al., in the study "Procedure for Determining the Biodegradation of Radiolabeled Substrates in a Calcareous Soil" https://doi.org/10.2136/sssaj2001.652347x indicated that some $CO_2$ fixation could indeed be fixed by $CaCO_3$, but it only happens in soils that are very carbonitic (>20% $CaCO_3$ by weight). In our study, some $CO_2$ fixation could occur in our Vertisol, however, the trends will all still be relative to the control.

Some references in which this method was used with calcareous soils and with soils that have a considerable content in carbonates are listed here:

Jones, D.L., Olivera-Ardid, S., Klumpp, E., Knief, C., Hill, P.W., Lehndorff, E., Bol, R., 2018. Moisture activation and carbon use efficiency of soil microbial communities along an aridity

gradient in the Atacama Desert. Soil Biology and Biochemistry 117, 68–71. https://doi.org/10.1016/j.soilbio.2017.10.026

Sánchez-Rodríguez, A.R., del Campillo, M.C., Torrent, J., Cooledge, E.C., Chadwick, D.R., Jones, D.L., 2024. Phosphorus fertilization promotes carbon cycling and negatively affects microbial carbon use efficiency in agricultural soils: Laboratory incubation experiments. Geoderma 450, 117038. https://doi.org/10.1016/j.geoderma.2024.117038

Sánchez-Rodríguez, A.R., Del Campillo, M.C., Torrent, J., Jones, D.L., 2014. Organic acids alleviate iron chlorosis in chickpea grown on two p-fertilized soils. J. Soil Sci. Plant Nutr. 35–46. https://doi.org/10.4067/S0718-95162014005000024

Finally, we have not tested the bioamendments with dead soil (autoclaved soil) because we wanted to develop the experiments under more realistic conditions. For that reason, we did not sterilise the soils used in our experiments.

**For readers not familiar with CUE, the calculation should be given. I**

**Response:** Following these comments from Reviewer#1, we have modified the Material and Methods section as follows, including some references:

"Microbial immobilisation of the 14C-substrate ($^{14}C_{imm}$) after 27 days was estimated as follows:

$$^{14}C_{imm} = {}^{14}C_{tot} - {}^{14}C_{NaCl} - {}^{14}CO_{0-27\ days} \qquad (1)$$

where $^{14}C_{tot}$ is the total amount of $^{14}C$-substrate added to the soil at time (t) = 0, $^{14}C_{NaCl}$ is the amount of $^{14}C$ recovered from the soil in the 1 M NaCl extracts at the end of the experiments and $_{14}CO_{0-27\ days}$ is the total amount of $^{14}C$ recovered as $^{14}CO_2$ during the experiments. Following Jones et al. (2018a,b), microbial CUE for the C substrate was then estimated as follows:

$$CUE = {}^{14}C_{imm} / ({}^{14}C_{imm} + {}^{14}CO_{0-27\ days}) \qquad (2)$$

**Is it relevant that the microbial life has not been determined?**

**Response:** We agree with Reviewer#1 here that the evaluation of soil microbial biomass could be interesting for understanding to correlate the changes observed in our study with soil microbial structure. However, our main concern in this study was to determine and highlight the positive effect that could have some of the tested bioamendments on soil heat resistance

and microbial CUE, regardless of the type of microorganisms. However, we will keep that comment in mind for future work.

**Minor comments**

**Title: I guess "resistance" is an interpretation of the CUE at various circumstances. So the title should be more like …..resistance derived from microbial CUE measurements…**

**Response:** We appreciate your suggestions; the title was modified as follows:

"Do composted bioamendments enhance Mediterranean agricultural soils resistance derived from microbial carbon use efficiency measurements to extreme heat-stress events?"

**Writing only about western Mediterranean region is strange, it also seems relevant for other regions in the world.**

**Response**: We totally agree with Reviewer#1 here, this work could be very useful for other areas with similar conditions. We have modified this part to suit more regions with similar soil and climate conditions:

"Summer temperatures in regions with Mediterranean semi-arid climate has reached high records and are expected to become more intense over the next few decades (Tejedor et al., 2024). The increase of global warming affects soil microorganisms and their activity in this region (Bañeras et al., 2022; Bérard et al., 2011), which involves changes in their structure and functioning, hence, altering nutrient cycling at a regional and global scale (Frey et al., 2013; Mooshammer et al., 2017; Reichstein et al., 2013)."

**53-53. This does not seem relevant for this manuscript.**

**Response:** Thank you for your observation. We have removed it to improve clearness of the Introduction.

**59 "poor" suggests that you have a opinion, which is strange if it is a natural state. I guess you mean low.**

**Response:** "Poor" was deleted according to this suggestion and "Low" was used in this occasion.

**61"negative impact", similar comment. Being calcareous is a state, it is not intervention which has an impact: it is the just the state of these soils which is problematic for certain crops.**

**Line 61:** Thanks for that, "negative impact" was removed and we used "which may constrain plant productivity" instead.

**59-63 I do not agree with the statement: "…. that calcareous soil … often lead to a negative impact on soil fertility and plant productivity". Impact is a strange word for a state of a soil, but also the idea that calcareous soils give low plant productivity is not correct: there is potential for negative effects. However, the productivity of calcareous soils is high when sufficient water and nutrient are applied (as is true for most soils). Maybe you mean alkaline soils, or the vertisols, when you want to talk about problematic soils.**

**Response:** We agree with Reviewer#1, we re-wrote the paragraph to be more precise:

"Calcareous soils are common in these areas, and they associated with limited availability of certain nutrients (e.g., P, Zn, Fe), which may constrain plant productivity under specific conditions"

**63 50 degrees Celsius is not that high for a barren dry arable soil. On a global scale many parts of the land have higher temperatures than air temperature. https://doi.org/10.1029/2010JG001486**

**Response:** We appreciate this observation and agree that surface soil temperatures under direct solar radiation in dry, exposed environments can indeed exceed 50 °C, often significantly higher than ambient air temperatures. We have modified this part of the Introduction section accordingly and included this useful reference:

"In addition, surface soil temperatures in arid dry agricultural fields often surpass air temperatures in such regions, and soil temperature exceeds 50 °C during extreme heat events, especially under intense solar radiation and low moisture conditions, which are common in semi-arid Mediterranean climates (Hamdi et al., 2011; Perkins, 2015; Vogel et al., 2017, Mildrexler et al., 2011)."

**70 Many authors have studied organic matter mineralisation, for example Kirby et al. showed an effect of nutrient on CO2 loss in a incubation experiment. https://doi.org/10.1016/j.soilbio.2013.09.032. In this paper the focus is on CUE. Please introduce this specific aspect. Why is it better or different?**

**Line 70:** Thanks Reviewer#1 for the valuable suggestions, modifications were done here to include this mentioned study:

"Microbial CUE is defined as the proportion of carbon taken up by microbes that is dedicated to growth rather than lost as $CO_2$ via soil respiration. It is a central determinant of soil carbon retention, and, therefore, a more meaningful indicator of microbial functioning and soil carbon dynamics than respiration alone (Allison et al., 2010; Ghee et al., 2013; Li et al., 2019; Mganga et al., 2022). According to Kirby et al., (2014), soil carbon sequestration and $CO_2$ released by soil microbes are directly affected by soil nutrients content. Other studies indicated that microbial CUE strongly influences soil C sequestration and is sensitive to various biotic (e.g., competition between species) and abiotic (e.g., pH, temperature) factors (Iven et al., 2023; Jones et al., 2019). This motivates research to identify which management strategies may enhance the resilience of semi-arid Mediterranean soils to extreme heat-stress events (> 40 °C) to prevent widespread soil degradation (Ferreira et al., 2022)."

**77 "most", do you mean "these"? Or do you mean other studies, then you should mention the other studies.**

**Response:** Expression modified (now we used "these studies" as we mentioned two studies here). Thanks.

**78-79 "such as calcareous soil … in regions?" Why not directly mention that vertisols and alfisols are specifically challenging?**

**Response:** Thanks for the appreciation, but we do not want to focus only on Vertisols and Inceptisols here, in this part of the Introduction, as other soil orders, Alfisols and Entisols, are also important in these regions. In the objectives, we focused on Vertisols and Inceptisols and explain the reasons why we chose them.

**81 and 84 "such as available P, further potentially deceasing CUE……23-24% reduction in CUE due to DAP and SSP.." Both sentences do not agree with each other. Does additional P increase or decrease CUE or do you mean that it is probably more complex?**

**Response:** Many thanks for your comment, we agree that this part of the Introduction was confusing. We would like to mention that CUE is negatively affected by the addition of mineral P fertilisers, according to previous research (by Sánchez-Rodríguez et al., 2024) and that, oppositely, Su et al. (2025) stated that CUE increased with the addition of organic fertilisers at

the same time as increased P availability to illustrate the need for more research in this topic. This shows the complexity of the relationship between P application to the soil and CUE as Reviewer#1 mentions. We have rewritten this part of the Introduction section to improve clarity and precision:

"While P is essential for microbial growth, inorganic P fertilisers reduced microbial CUE in the short-term, as stated by Sánchez-Rodríguez et al. (2024). They found a significant reduction in microbial CUE (23-24%) in typical Mediterranean soils (Inceptisol, Alfisol, and Vertisol) when P was applied to the soil as diammonium phosphate or single superphosphate, which may be due to shifts in microbial community and nutrient dynamics or stoichiometric imbalances. However, Su et al. (2025) stated that the addition of organic fertilisers significantly enhanced CUE at the same time as increased P availability. These studies are fundamental to design sustainable strategies in which the adoption of organic agriculture practices, such as compost application, to obtain more resilient farming systems (Moreno-Pérez, 2023) in line with European policies and strategies (Rato-Nunes et al., 2017). However, the effects on microbial CUE after the application of organic amendments under extreme heat stress is still not clear."

**97 "little is known about their impact on CUE" Is that true, not at a first glance. https://doi.org/10.1186/s13213-024-01780-9, https://doi.org/10.1007/s42832-022-0137-3, https://doi.org/10.7717/peerj.12131, https://doi.org/10.1016/j.soilbio.2024.109531. So, please be more precise.**

**Response:** Thanks for the references. We agree with Reviewer#1, previous research has deal with the impact of extreme heat on soil microbial CUE. However, there is not much work on microbial CUE and organic amendments under heat stress in soils from Mediterranean regions. We revised the stated section to suit better the idea and to specify the knowledge gap that we are working on:

"Although previous research has evaluated microbial CUE under extreme heat scenarios (e.g., Dang et al., 2024; Zhang et al., 2022, Adingo et al., 2021), there is a critical knowledge gap in evaluating the impact of bioamendments application to the soil on microbial CUE under severe climatic conditions as extreme heat waves, such as those documented in Mediterranean areas. This information is needed to design holistic strategies that include the use and potential benefits of bioamendments in Mediterranean regions with challenging soil properties (low organic matter content and reduced availability of P) that are subjected to extreme heat-stress events."

**103 Earlier you mentioned that the calcareous soils were problematic for phosphorous, and now you include a non-calcareous soil. Please explain you choices: for example you chose soils with a low P availability.**

**Line 103**: Effectively, both soils have a low content in P. However, the choice of the calcareous soil (Vertisol) was for its content in carbonate, which limits nutrients availability for plants, and also this type of soil is typical in the area of the study that normally suffers from successive and continuous heat stress. The Inceptisol is other typical soil in the area of the study with low P availability too. The use of these soils helps us have an idea on the behaviour of soil microorganisms ($CO_2$ respiration and microbial CUE) of these two contrasting soils (calcareous and no calcareous soils) under heat stress with the application of different bioamendments.

We have clarified the idea behind the decision of using the mentioned soils on the manuscript:

"This study investigated the effects of various bioamendments (composted olive mill pomace, composted biosolids, and composted solid urban residue) and a mineral fertiliser (diammonium phosphate) on microbial CUE and key soil chemical properties (including available phosphorus, carbon, nitrogen, pH) in a calcareous Vertisol and a non-calcareous Inceptisol, both collected from Mediterranean regions and with a low P availability. These two soils exhibit contrasting carbonate and clay contents, enabling the assessment of whether calcareous soils buffer the effects of extreme heat stress, and the role of added bioamendments in increasing their resistance."

**104 "soil biogeochemistry". Please write more precise: you do not study soil biogeochemistry, you have determined P-Olsen and extractable N.**

**Line 104:** We appreciate this comment. Please, see our response to the previous comment as the modifications are included there.

**106 "than conventional fertiliser". This does not seem a fair comparison. If so, then you should add similar amount of nutrients using various mineral fertilisers, including micronutrient. Conventional fertiliser is not a very good term: in many countries this is animal manure, in other mineral fertiliser. So mineral fertiliser is a better term. In most studies authors use a simple fertilisation advice for the soils, and choosing different fertilisers. In the current research you might have deficiencies for N, Mg, S etc. By using**

**soils from farms, you probably use well fertilised soils, without deficiencies, at least not in micronutrients.**

**Line 106:** We appreciate this suggestion done by Reviewer#1. "Conventional fertiliser" has been modified by "mineral fertiliser" through all the manuscript to increase clarity. Thank you for this insightful comment. Regarding the comparison between mineral fertiliser (diammonium phosphate) and bioamendments: we agree that a direct comparison is only meaningful when both are applied with the same agronomic objective, in this case, enhancing phosphorus availability. In our study, all treatments were applied at rational rates that reflect typical field practices aimed at improving soil P availability (but not the rest of elements). Under this framework, the comparison can be considered fair, as both the mineral fertiliser and bioamendments were used to serve the same function in the short-term (increasing soil P by introducing equal amounts of P by all the used treatments, mineral fertiliser or bioamendments), and the results reflect the relative efficacy of these treatments under the same experimental conditions.

**108 "a more buffered". This is in contrast to line 79 were you state that the vertisols are prone to high moisture and reduced oxygen.**

**Line 108**: Thank you for your observation. We agree that the phrasing may have caused confusion. The statement referring to Vertisols as "*more buffered*" was meant to describe their thermal and pH buffering capacity, due to their high clay and carbonate content, rather than suggesting they are universally buffered against all environmental stresses. In contrast, the earlier mention highlights that Vertisols are prone to poor drainage, which can lead to high moisture content and low oxygen availability, particularly under wet conditions. These characteristics can indeed reduce microbial CUE under certain circumstances.

We have revised the text to clarify the specific buffering properties being referred to and to avoid generalizations that may imply inconsistency.

**145-150 Did you make batches of soil+amendment mixtures, and did you sample these mixtures for experiment 1-3? The text does not explain how you did this. This is relevant as these amendments have a structure (compost contain large particles, and the phosphate minerals are also particles). If you sample 2,5 gram soil (mixture of soil+amendment), then the samples are probably heterogeneous. This might explain the large variance in figure 2. Also in figure 3 there is a large variation for soluble N in the**

**treatment with DAP although in every soil you have added the same amount of N+P. It is rather problematic that the variation is so large, when you expect very similar results.**

**Lines 145-150:** We thank Reviewer#1 for this important observation. We have now clarified in the Methods section that soil and amendment mixtures were thoroughly homogenized before subsampling, and that all replicates were prepared identically. The higher variability in soluble N in DAP-treated soils could be due to the small amount of this fertiliser added to each soil sample (the smallest considering all the treatments as its P concentration was the highest); however, although a high variability was observed with DAP in extractable total N, this treatment produced the highest values of this variable.

We add the modifications done in this part of Material and Methods as follows:

"To ensure representative amendment application, larger bulk mixtures of soil and treatments than needed were prepared. Previously, the fertiliser and the bioamendments were ground and sieved to 0.5 mm to ensure homogeneity. The required amount of fertiliser or bioamendment needed to reach the target P level of 50 mg kg$^{-1}$ in 2.5 g of soil was extremely small and technically challenging to apply accurately. Therefore, soil was gradually added to pre-weighed fertiliser or bioamendment in larger batches, followed by thorough mixing to ensure homogeneity. From each homogenised batch, subsamples of each mixture (fertiliser / bioamendment and soil) were used for incubation experiments. The obtained mixtures were wetted as explained in each experiment to ca. 0.18 g g-1 gravimetric moisture content (43 and 49 % water-filled pore space for the Vertisol and the Inceptisol, respectively) to activate soil microorganisms and reflect the relatively dry conditions typical of Mediterranean topsoil during summer heatwaves."

**150 It seems like a small amount of water: 0.18 gram water per gram dry material? Normally soils are wetted until a certain percentage of water filled pore space: ±70% of the maximum, to have a good circumstances for plant roots.**

**Line 150**: many thanks for your comment. The amount of water added to our soils (gravimetric moisture of 0.18 g g$^{-1}$; 0.45 g per 2.5 g of soil) corresponds to approximately 43-49% of the water-filled pore space for the Vertisol and the Inceptisol, respectively. This moisture level was selected to maintain microbial activity, prevent anaerobic conditions during incubation, and reflect the conditions typical of Mediterranean topsoil when P fertiliser of bioamendments are applied. We wanted to simulate the conditions that can be found in these soils when the P

fertilisers or bioamendments are added to the soil before sowing cereal (durum wheat, typical in this region).

**190 Soils seem deficient in zinc according to these measurements.**

**Line 190:** Indeed, both soils are deficient in Zn. This is normal in some soils collected from Mediterranean areas. In this case, we think this is not important for our study as we are focused on soil respiration and CUE under the conditions that are common in these soils just when the P fertilizer or bioamendments are applied and not in plants (critical soil Zn values are calculated as indicators for plants).

**197 data in table 2 are strange for "volatile solid content" or "oxidable organic carbon". How is it possible that the volatile solid content (proxi for organic matter, and water to clays) is lower than organic carbon? You would expect a factor 2 between both.**

**Line 197 :** Thank you very much for pointing this out, after reviewing the data and doing new measurements, we confirmed that there was a mistake in the oxidizable carbon values reported in Table 2. Specifically, a data entry error occurred during formatting, leading to incorrect values that did not reflect the actual measurements. We have now corrected the values, which are consistent with expectations. This correction does not affect the interpretation of our results or conclusions, as the trends remain unchanged. The updated values and clarification of units have been incorporated in the revised Table 2 and its text.

**200 Can $^{14}CO_2$ might also be precipitated as CaCO3? For example:. https://doi.org/10.1016/S0168-1923(02)00231-9. So how sure are you of the measurements? Has it been tested in dead soil?**

**Response:** Firstly, we replied to this question in a more general way in "Specific comments, I wonder if the method is correct. CO2 can be precipitated by Ca and Mg rich materials at certain CO2 concentrations. The vertisols but also the composts probably contain carbonates. Should this be tested with a dead soil?" Please, read our pecious response.

Additionally, we added some references of studies in which microbial CUE was calculated in calcareous soils, and this method was useful to understand the reactions occurring in these soils in relation with C and nutrient dynamics in soil.

In this study, we did not use sterile (dead) soils as a control. Our experimental design aimed to simulate realistic conditions, including the native microbial communities of each soil type to

assess how bioamendments affect microbial carbon use efficiency (CUE) under heat stress. Given the one-week incubation period, we expected microbial activity to remain stable making the use of sterile soils less representative of real conditions. We acknowledge, however, that $^{14}CO_2$ precipitation as $CaCO_3$ is a potential limitation, particularly in calcareous soils. While no significant anomalies were observed in comparison to the more acidic Inceptisol, we recognize the need for further studies using sterilized controls to directly quantify $CO_2$ precipitation and improve CUE estimations in carbonate-rich soils. This point has been added to the discussion as a methodological limitation in the section 4.2 "*Wider implications and priorities for future research*":

"Furthermore, we recognize a potential methodological limitation related to the use of $^{14}C$-labelled glucose to calculate microbial CUE in calcareous soils. Specifically, the potential precipitation of $^{14}CO_2$ as $CaCO_3$ may have led to an underestimation of $CO_2$ emissions from soil and overestimation of CUE. While no major anomalies were observed between the calcareous and the non-calcareous soil, future studies should consider including sterilised soil controls to assess this effect more accurately and enhance the robustness of CUE calculations in carbonate-rich systems."

**209, 238, 250: Experiment 1, 2 and 3 have been performed in different tubes:**

**1: 2.5 gram soil, 0.2 ml water, 50 ml tube, 0.25 ml labelled glucose.**

**2: 2.5 gram soil, 0.2 ml water, in a 1,5 ml (?) Eppendorf tube. How does this fit?**

**3:2.5 gram soil, 0.2 ml water in 50 ml tube, 0.25 ml labelled glucose.**

**Unclear are the effects of the differences. Unclear: are the tubes closed from air, or open? Does the soil dry out during the 27 days of having a 1 M NaOH trap on top of it?**

**I would not use the word "soil" here, when you mean a mixture of soil+amendment. A reader expects that you add the amendment afterwards when he/she reads "soil".**

**Response:** We appreciate this comment. The used flask volume has been corrected in the second experiment. It was the same volume for the three experiments as Reviewer#1 pointed out. The modifications were done only in the second experiment (2.5.2), as follows:

"To determine changes in chemical characteristics in soil receiving bioamendments or inorganic fertiliser after an extreme heat stress event, other experiment was built in parallel to Experiment 1, with 2.5 g of soil ($n = 3$ per treatment, soil type and temperature) placed in a 50

ml polypropylene centrifuge tube, rewetted with 200 µl of DI H$_2$O, and pre-incubated at 20 °C for 1 week."

Concerning the Falcon tubes, they were hermetically closed during the experiment as mentioned. This prevent or minimises soil drying:

"A 6 ml polypropylene vial containing 1 ml of 1 M NaOH was then placed above the soil surface to capture respired $^{14}CO^2$ and the tubes sealed".

Finally, in each subsection (2.5.1, 2.5.2 and 2.5.3) we have used "soil or soil and fertiliser or amendment mixture" according to Reviewer#1 comment.

**2.5.2: In experiment 2: pH, mineral N, and P Olsen were determined. Did you do this on this small 2.5 gram sample? How?**

**Rather unclear how you derive CUE.**

**Response: Yes, we did all the analysis with 2.5 g of soil. We have slightly modified this section to include that information:**

"Then, soil pH and EC were determined on 0.5 g of fresh soil or soil and fertiliser or amendment mixture following a 1:2.5 w/v (soil:solution) DI H$_2$O extraction using micro-pH meter and micro-conductivity meter. Gravimetric soil moisture content was determined by drying 0.5 g of soil at 105 °C for 24 h. Total extractable N and organic C were analysed on 0.5 g of soil using 0.5 M K$_2$SO$_4$ extracts (1:5 w/v) on a Multi N/C 2100 S analyser (AnalytikJena, Jena, Germany). Soil ammonium (NH$_4^+$), nitrate (NO$_3^-$), were measured colorimetrically from the 0.5 M K$_2$SO$_4$ extracts using the methods described in Mulvaney (1996) and Miranda et al. (2001), respectively. Available phosphorus (Olsen-P) was determined on 1 g of soil, using the method of Olsen et al. (1954) as described previously."

**263-275 Please give calculation of CUE in the methods.**

**Response:** Done. The Material and methods section includes the equation and two references of the method as previously mentioned in "Specific comments". We do not add the same information here to avoid repetition.

**357-364 So nitrification is rather slow. One would expect that all NH4 would be transformed into NO3 after so many days, at least for DAP.**

**Response:** We agree with Reviewer#1. Nitrification was slow. that under optimal conditions, $NH_4^+$ from DAP is generally expected to be nitrified to $NO_3^-$ over a period of several days. Different limitations could have occurred:

(i)      High incubation temperatures (40 °C but especially 50 °C in the Vertisol) likely inhibited the activity of nitrifying microorganisms, which are known to be heat-sensitive and function optimally below 35-40 °C.

(ii)     The relatively short incubation period (9 days) may not have allowed complete nitrification to occur under the conditions of the experiments.

**425 "not many", if so then you should mention these few studies.**

**Response:** Corrected to mention that there are not studies:

"Moreover, there is a lack of studies that evaluate the role of soil type and different amendments in buffering the negative effect of extreme heat events on CUE, especially, when soils are exposed to extreme heat stress where air temperatures exceed 40 °C…"

**440 Do you add microbial live to a sample by adding compost? Or is that negligible to soil?**

**Response:** Yes, microbial life is added when compost is applied to the soil, and the microbial community of the soil could be modified. However, the assessment of the effect on microbial CUE after compost application under different temperatures (including extreme temperatures) is the aim of the study.

**445 Strange. You Spanish soils have of course been adapted to 50°C. You have given the temperatures, and temperatures of soils are often much higher than air temperatures. Otherwise you should have chosen soils from Scandinavia or some other region without warm summers/sun.**

**Response:** We agree with Reviewer#1; topsoil temperatures often exceed air temperatures, particularly under direct solar radiation in arid and semi-arid environments. However, in our study, we included a range of temperatures to evaluate the potential shift in microbial $CO_2$ release from the soil and microbial CUE in these soils (that occurred at 50 °C in both soils). So, we could say that our soil microbial communities (in the vertisol and the inceptisol used here) are resistant to temperatures up to 40 °C but a dramatical decrease in microbial CUE occurred at 50 °C, except when composted olive mill pomace was added to these soils (when

the decrease in microbial CUE was not as evident as in the other treatments, including control, DAP and other bioamendments).

**498 "…. Especially DAP, lead to … NH4… this reflects either increased mineralization of organic N or….". That seems to make no sense. You add NH4 with DAP (NH4 and HPO4), so you do not need a biological process to find NH4.**

**Response:** We have improved this phrase as we agree with Reviewer#1:

"According to previous research (Dai et al., 2020), the application of fertilisers, especially diammonium phosphate, leads to a greater accumulation of $NH_4^+$ (Fig. S5) and dissolved TN due to the high amount of $NH_4^+$ that this fertiliser provides to the soil."

**528 "vulnerability". I wonder if you can state this on the basis of your two soils**

**Response:** "Done:

"This study highlights the vulnerability of Vertisols and Inceptisols located in semi-arid regions to extreme heat-stress events."

**570-575. That is a rather unrealistic conclusion: the availability of compost per hectare in the EU is very small compared to potential need. It is probably much easier to keep the soil covered with crops, being cover crops or crops.**

**Response:** We partially agree here with reviewer#1. The conclusions drawn in this study are based specifically on the observed effects of a set of bioamendments that are commonly used in agriculture Southern Europe. Since the study did not include any specific cropping systems or crop cover strategies, we agree that it is not possible to make broader comparisons between bioamendment application and alternative approaches such as maintaining permanent soil cover. We acknowledge that future research would benefit from directly comparing the effectiveness of different soil management strategies, such as bioamendment application versus cover cropping, in enhancing soil resilience to heat stress; and we agree with Reviewer#1 in that cover crops could be a potential option for these vulnerable soils to extreme heat events located in Mediterranean regions.

---

## Author Comment (AC2)

**General**

**This study investigates the effect of bioamendments and heat stress in two soil types on microbial respiration and carbon use efficiency. This is a very relevant topic and fits well the scope of SOIL. I have, however, strong reservations about the quality of the study.**

**Response:** We appreciate the time and the effort Reviewer#2 invested in the review our study, and we hope to have improved the quality of our work in our new version.

**Reviewer: Overall, I find the introduction and methods hard to read. The introduction is overall a bit confused and does not clearly frame the interplay between heat stress and amendments in affecting the carbon cycle.**

**Response:** Many thanks for your valuable comment. We have modified the Introduction and Material and Methods sections to improve their clarity, to be more appropriate for the research subject and conclusions. Please, see our responses to the next questions / comments and also our line-by-line responses.

Also, the relationship between bioamendments and carbon cycling was added in the Introduction section:

*"Bioamendments may improve agricultural productivity by modifying soil microbial community composition and activity (Kok et al., 2023), enhancing extracellular enzymatic production for nutrient utilisation and microbial acclimation (Conant et al., 2011), providing energy and essential nutrients for soil microorganisms (C, N and P; Wang and Kuyakov, 2023), and influencing soil biogeochemistry (Mooshammer et al., 2017). They also promote improvements in soil structure and reduce soil bulk density, which could result in plant growth enhancement. Although previous research has evaluated microbial CUE under extreme heat scenarios (e.g., Dang et al., 2024; Zhang et al., 2022, Adingo et al., 2021), there is a critical knowledge gap in evaluating the impact of bioamendments application to the soil on microbial CUE under severe climatic conditions as extreme heat waves, such as those documented in Mediterranean areas. This information is needed to design holistic strategies that include the use and potential benefits of bioamendments in Mediterranean regions with challenging soil properties (low organic matter content and reduced availability of P) that are subjected to extreme heat-stress events.»*

Please, see the new version of the manuscript where all the changes are shown (especially in the version with track changes).

**Reviewer: there is no clear or accurate description of what CUE is, how it is conceptualised across scales and also methods, and finally more precisely how it contributes to ecosystem C cycling.**

**Response:** We appreciate your comment. We have modified our manuscript and included your suggestions in the Introduction section, paying special attention to the definition and the role of CUE and how it directly affects ecosystem C cycling according to this suggestion:

*"The alteration in soil microbial communities exposed to extreme heat stress often induces perturbation in organic matter mineralisation and C sequestration. Moreover, previous studies (Yang et al., 2023; Beugnon et al., 2025) have shown that heat stress reduces soil microbial C use efficiency (CUE; the proportion of C assimilated by soil microorganisms and allocated to biomass production rather than released as $CO_2$ through soil respiration), while simultaneously increasing respiration rates. This, in turn, can enhance organic matter mineralisation. In addition, soil C stocks depletion could dramatically occur under this situation if the soil does not receive sufficient C inputs. Microbial CUE plays a key role in soil C retention, soil organic C (SOC) storage and its global spatial variation, since it represents a dual microbial control point over both SOC accumulation (via biomass production) and SOC loss (via respiration; Tao et al., 2023). Therefore, microbial CUE is a more meaningful indicator of soil microbial functioning and C dynamics than respiration alone (Allison et al., 2010; Ghee et al., 2013; Li et al., 2019; Mganga et al., 2022). Additionally, soil C sequestration and $CO_2$ released by soil microbes are directly affected by soil nutrients content according to Kirby et al. (2014). Moreover, other research support that microbial CUE strongly influences soil C sequestration and is sensitive to various biotic (e.g., competition between species) and abiotic (e.g., pH, temperature) factors (Iven et al., 2023; Tao et al., 2023; Jones et al., 2019) . However, there is a gap of knowledge related to soil management strategies (bioamendments application) that may enhance the resilience of semi-arid soils to extreme heat-stress events (> 40 °C) to prevent widespread soil degradation (Ferreira et al., 2022). »*

**Reviewer: The method section is also overall unclear because some elements are mentioned in passing before being explained. Quite a few methodological details are missing (as listed in the detailed comments below), including the calculation method for CUE.**

**Response:** The Material and Methods section was improved to be clearer and easier for reproduction (see our responses to your line-by-line comments). We have included in supplementary file figure S4, which is a schematic overview of the experimental design to explain the used methods as clearer as possible. Also, the calculation of CUE has been added for more clarity.

*"Additionally, microbial immobilisation of the $^{14}C$-substrate ($^{14}C_{imm}$) after the monitoring period was estimated as follows:*

$$^{14}C_{imm} = {}^{14}C_{tot} - {}^{14}C_{NaCl} - {}^{14}CO_{0-t\ days} \qquad (1)$$

*where $^{14}C_{tot}$ is the total amount of $^{14}C$-substrate added to the soil, $^{14}C_{NaCl}$ is the amount of $^{14}C$ recovered from the soil in the 1 M NaCl extracts at the end of the experiments and $^{14}CO_{0-t\ days}$ is the total amount of $^{14}C$ recovered as $^{14}CO_2$ during the experiments (21-27 days). Then, microbial CUE for the $^{14}C$ substrate was estimated as follows following Jones et al. (2018a,b):*

$$microbial\ CUE = {}^{14}C_{imm} / ({}^{14}C_{imm} + {}^{14}CO_{0-t\ days}) \qquad (2)\ »$$

**Reviewer: Most importantly, I have concerns about the validity of the method used. The choice of 14C glucose addition is interesting here as a standardised assay to quantify CUE, since the treatments include amendments containing carbon in different forms (different C:N), whose incorporation into microbial biomass probably differ from that of glucose. This choice could potentially be justified, but it needs careful explanation and a clear description of what can be concluded about the c cycle from it in the context of this study with different amendments, both in introduction and discussion.**

**Response:** We thank Reviewer#2 for highlighting this important methodological point. We agree that the bioamendments used in our study contain carbon in complex and heterogeneous forms (different C:N ratios, chemical structures, and decomposition rates), which would be assimilated differently into microbial biomass compared to a simple substrate like glucose. Our rationale for using $^{14}C$-labelled glucose was to provide a standardised, labile carbon source across all treatments and soil types, in order to assess microbial CUE under different thermal stress conditions. By using the same substrate (glucose), we were able to directly compare microbial C allocation strategies (respiration vs. assimilation) across soils and amendments, without confounding effects from differences in the intrinsic quality or bioavailability of amendment-derived C. This approach follows established protocols for assessing microbial CUE under controlled conditions (e.g., Jones et al., 2019; Glanville et al., 2016; Sánchez-Rodríguez et al., 2024). Thus, while our results cannot be taken to describe the complete fate of amendment-derived carbon, they do provide valuable insights into how bioamendments influence microbial resilience and efficiency under extreme heat stress when microbes are supplied with labile carbon.

In the literature there are multiple references in which this method is used. We have added a few of them:

Jones, D.L., Olivera-Ardid, S., Klumpp, E., Knief, C., Hill, P.W., Lehndorff, E., Bol, R., 2018. Moisture activation and carbon use efficiency of soil microbial communities along an aridity gradient in the Atacama Desert. Soil Biology and Biochemistry 117, 68–71. https://doi.org/10.1016/j.soilbio.2017.10.026

Jones, D. L., Cooledge, E. C., Hoyle, F. C., Griffiths, R. I., and Murphy, D. V.: pH and exchangeable aluminumaluminium are major regulators of microbial energy flow and carbon use efficiency in soil microbial communities, Soil Biol. Biochem., 138, doi:10.1016/j.soilbio.2019.107584, 2019.

Sánchez-Rodríguez, A.R., del Campillo, M.C., Torrent, J., Cooledge, E.C., Chadwick, D.R., Jones, D.L., 2024. Phosphorus fertilization promotes carbon cycling and negatively affects microbial carbon use efficiency in agricultural soils: Laboratory incubation experiments. Geoderma 450, 117038. https://doi.org/10.1016/j.geoderma.2024.117038

Sánchez-Rodríguez, A.R., Del Campillo, M.C., Torrent, J., Jones, D.L., 2014. Organic acids alleviate iron chlorosis in chickpea grown on two p-fertilized soils. J. Soil Sci. Plant Nutr. 35–46. https://doi.org/10.4067/S0718-95162014005000024

Glanville, H. C., Hill, P. W., Schnepf, A., Oburger, E., and Jones, D. L.: Combined use of empirical data and mathematical modelling to better estimate the microbial turnover of isotopically labelled carbon substrates in soil. Soil Biology and Biochemistry 94, 154-168, doi:https://doi.org/10.1016/j.soilbio.2015.11.016, 2016.

**Reviewer: If 14C glucose as a general method could be, perhaps, justified, different incubation times for different treatments constitutes a methodological bias. It is clearly stated that different treatments were subject to different incubation times. Incubation time (after which 14C remaining into the soil was measured, which I assume was used to estimate incorporation of 14C into microbial biomass) appears to be based on the time it takes for CO2 emission rates to stabilised, which expectedly differed between temperature treatments. In my sense, this does not allow comparison of CUE in the different temperature treatments, thus providing a biased method to address the key question of understanding the impact of heat stress on CUE. This is because incubation time in substrate incorporation methods to calculate CUE determines largely how CUE can be conceptualised, with increasing incubation time increasing the chances of added inputs being exuded, turned-over or maintenance respiration, rather than contributing to growth. If not accounted for, these processes can lead to overestimations of the fraction of substrate assimilated into from the classical equation:**

**CUE = 14C biomass / (14C biomass + 14 respired).**

**Response:** We appreciate the reviewer's insightful comment regarding incubation time. We agree that incubation time is a critical factor influencing microbial CUE because longer durations may incorporate processes beyond initial assimilation (e.g., turnover, maintenance respiration).

Our approach was guided to capture complete respiration dynamics after 14C glucose addition. Microbial respiration was stabilised at different times depending on the temperature. We, therefore, extended incubations until $^{14}CO_2$ release plateaued for each temperature, following established protocols (Jones et al., 2019; Glanville et al., 2016). This ensured that cumulative $CO_2$ release reflected the full mineralisation of added glucose under each thermal regime with the different bioamendments. Regarding the comparability across soils/treatments. If a fixed length of monitoring of the experiment had been applied (e.g., 21 or 27 days), the faster dynamics at high temperature would have been truncated, underestimating respiration and overestimating CUE relative to lower temperatures (more details at line-by-line answers below). This only happened for Experiment 1 and not Experiments 2 or 3 (please, see Fig. S4 added to clarify our methodology and this point).

**Reviewer: Due to this lack of clear framing, and particularly of partly inadequate methodology, I cannot recommend publication.**

**Response:** We thank Reviewer#2 for this critical assessment. We recognise that in the previous submitted version the framing of CUE and the methodological description were not sufficiently clear. However, we have substantially improved the manuscript to address these points. These revisions strengthen the framing and clarify the Introduction and Methodology sections beside adding a schematic overview of the experimental design in supplementary file (Fig. S4, added to clarify our experimental design for each experiment). Thus, while we acknowledge the limitations of our study (we have detailed them in the Discussion section), we think that the revised manuscript provides robust and relevant insights into the resilience of Mediterranean soils under extreme heat stress and the effects of bioamendments (scientific literature is lacking or limited related to this topic). Moreover, please see our response to your previous comment.

**Reviewer: I recommend reading Geyer et al. (2016) (DOI:10.1007/s10533-016-0191-y) to shed light on how incubation times impacts not only results, but also conceptualisation of CUE.**

**Response:** Many thanks for the suggested reference. However, this study uses other methods for CUE calculation depending on the scale that is assessed. Nevertheless, although we recognize that these methods could be a valuable add to our study (as mentioned in the manuscript limitations section), we have used other method to quantify microbial CUE to evaluate the effects of the different bioamendments in two soils with contrasting properties. The method that we have used is widely used in the scientific literature as we mentioned in our previous responses (adding more references to support the use of this method to achieve our main objectives).

**Detailed comments:**

**Reviewer: Can only Line 50-53; Line 6: Syntax errors**

Response: We appreciate this comment, but we do not understand this comment to line 6:

"ᵃThese authors contributed equally to the manuscript and are considered co-first authors."

**Reviewer: L66-70: The formulations are a bit inaccurate here, and I have issues with the concepts. 1. "Consequently", line 67; the death or dormancy, and the change in composition are responses of the community that partly define adaptation, not the cause of adaptation; death does not trigger a shift in metabolism, it IS a pretty dramatic shift in metabolism… high temperature is the cause of all that (death, dormancy, shift in metabolism and adaptation). 2. Also, "shift in metabolism to facilitate thermal adaptation"… I think a shift in metabolism is a form of adaptation (acclimation perhaps) itself, like species turnover and de novo genetic mutations**

Response: We agree with Reviewer#2, this sentence could be confusing. We have reformulated this paragraph to be clearer following these comments:

"This often surpasses the microbial thermal optimum, resulting in the death or dormancy of thermosensitive taxa (Donhauser et al., 2020; Riah-Anglet et al., 2015) and changes in the community composition (Bérard et al., 2011; Hawkes and Keitt, 2015). These responses–together with physiological adjustments within surviving taxa–constitute microbial thermal adaptation and could be accompanied by altered metabolic activity, resulting in elevated respiration in the remaining thermotolerant species (Anjileli et al., 2021; Bardgett and Caruso, 2020). The alteration in soil microbial communities exposed to extreme heat stress often induces perturbation in organic matter mineralisation and C sequestration. Moreover, previous studies (Yang et al., 2023; Beugnon et al., 2025) have shown that heat stress reduces soil microbial C use efficiency (CUE; the proportion of C assimilated by soil microorganisms and allocated to biomass production rather than released as $CO_2$ through soil respiration), while simultaneously increasing respiration rates. This, in turn, can enhance organic matter mineralisation. In addition, soil C stocks depletion could dramatically occur under this situation if the soil does not receive sufficient C inputs. Microbial CUE plays a key role in soil C retention, soil organic C (SOC) storage and its global spatial variation, since it represents a dual microbial control point over both SOC accumulation (via biomass production) and SOC loss (via respiration; Tao et al., 2023). Therefore, microbial CUE is a more meaningful indicator of soil microbial functioning and C dynamics than respiration alone (Allison et al., 2010; Ghee et al., 2013; Li et al., 2019; Mganga et al., 2022)."

**Reviewer: "inadvertently" is not the right word. increasing OM mineralisation, deleting soil C stocks and reducing CUE… it sounds like all those would be the direct consequence of an increased respiration in the remaining thermotolerant species. I think this is a large oversimplification. It needs to be laid out how increased mineralisation and decreased CUE may contribute to decrease C stocks, and in**

**which condition would this lead to a decrease in C stocks (with respect to plant C inputs particularly).**

**Response:** We partially agree with this comment. First, we have deleted "inadvertently" (please, see our response to your previous comment). Then, our study is limited to the effect of heat stress in bare soil (without plants). We are focused on capturing the reaction of soil microbes to common heat stress in the areas of the study and the effects of bioamendments under these conditions (for soil fertility and microbial CUE). Nevertheless, we agree that the introduction of other factors as carbon input by plants could affect the carbon stocks in soil, but this is not the aim of the study. We have modified this paragraph to be more precise (please, see this modified paragraph in our previous response).

**Reviewer: L70-74: how does the fact that CUE matters to C cycling and is sensitive to various factors justifies the need to understand resilience? We want to know specifically how resilience relates to soil C cycling, and how understanding CUE's response to drought and temperature is critical, because of this role in C cycling, to understand resilience…**

**Response:** We appreciate this comment to enhance the clarity of the Introduction. We have introduced the meaning of Microbial CUE and how it collaborates in soil C cyclin under drought:

*"…Moreover, previous studies (Yang et al., 2023; Beugnon et al., 2025) have shown that heat stress reduces soil microbial C use efficiency (CUE; the proportion of C assimilated by soil microorganisms and allocated to biomass production rather than released as $CO_2$ through soil respiration), while simultaneously increasing respiration rates. This, in turn, can enhance organic matter mineralisation. In addition, soil C stocks depletion could dramatically occur under this situation if the soil does not receive sufficient C inputs. Microbial CUE plays a key role in soil C retention, soil organic C (SOC) storage and its global spatial variation, since it represents a dual microbial control point over both SOC accumulation (via biomass production) and SOC loss (via respiration; Tao et al., 2023). Therefore, microbial CUE is a more meaningful indicator of soil microbial functioning and C dynamics than respiration alone (Allison et al., 2010; Ghee et al., 2013; Li et al., 2019; Mganga et al., 2022). Additionally, soil C sequestration and $CO_2$ released by soil microbes are directly affected by soil nutrients content according to Kirby et al. (2014). Moreover, other research support that microbial CUE strongly influences soil C sequestration and is sensitive to various biotic (e.g., competition between species) and abiotic (e.g., pH, temperature) factors (Iven et al., 2023; Tao et al., 2023; Jones et al., 2019) . However, there is a gap of knowledge related to soil management strategies*

*(bioamendments application) that may enhance the resilience of semi-arid soils to extreme heat-stress events (> 40 °C) to prevent widespread soil degradation (Ferreira et al., 2022)."*

Also, we have modified another paragraph in the Introduction section to describe the importance of evaluating microbial CUE under extreme heat stress and drought conditions:

*"Previous research has examined the impact of extreme heat waves on certain soils located in Mediterranean areas (Bañeras et al., 2022; Bérard et al., 2011). However, these studies do not assess the effects of extreme heat on these soils with challenging conditions for microbial CUE in arid or semi-arid regions, where periods of low moisture and aerobic conditions in soil are frequent, which tend to reduce microbial CUE (Zheng et al., 2019). Additionally, the presence of calcium carbonate in soils located in these regions limits microbial nutrient availability such as available phosphorus (P)."*

**Reviewer: L78: "challenging CUE conditions"… what are those? It was not mentioned before that calcareous soils have low CUE or why.**

Response: we appreciate your comment to clarify the meaning of "Challenging CUE conditions", we have added further explanation: please, see our response to your previous comment. We referred to arid-semi arid regions (low soil moisture) and considerable carbonates content in soil that limits P availability and affects microbial CUE (so, fertilization is another key factor).

Furthermore, we have added more details about microbial CUE and calcareous soils. Please, find our modifications as follows:

*"While P is essential for microbial growth, the application of inorganic P fertilisers could reduce microbial CUE in the short-term, as stated by Sánchez-Rodríguez et al. (2024). They found a significant decrease in microbial CUE (23–24%) in typical Mediterranean soils (Inceptisol, Alfisol, and Vertisol) when P was applied to the soil as diammonium phosphate or single superphosphate, which may be due to shifts in soil microbial community and nutrient dynamics or stoichiometric imbalances. However, Su et al. (2025) stated that the addition of organic fertilisers significantly enhanced CUE at the same time as increased P availability. These studies are fundamental for designing sustainable strategies that incorporate agricultural practices aligned with circular economy, such as compost application, to build more resilient farming systems (Moreno-Pérez, 2023) in accordance with European policies and strategies (Rato-Nunes et al., 2017). However, the effects on microbial CUE after the application of organic amendments under extreme heat stress is still not clear. "*

**Reviewer: L81: we need a ref to justify that low P availability would decrease CUE.**

**Response:** thank you for your comment. However, this sentence was removed following Reviewer's 1 recommendation.

**Reviewer: L82: compost application is a fairly common practice that is absolutely not unique to "organic agricultural practices".**

**Response:** Many thanks for your comment. This paragraph has been reformulated to be more precise according to this comment:

*"These studies are fundamental for designing sustainable strategies that incorporate agricultural practices aligned with circular economy, such as compost application, to build more resilient farming systems (Moreno-Pérez, 2023) in accordance with European policies and strategies (Rato-Nunes et al., 2017). However, the effects on microbial CUE after the application of organic amendments under extreme heat stress is still not clear."*

**Methods**

**Reviewer: L200-201: suddenly 14C is mentioned. I am not sure I understand here. The biosolids are obtained from commercial sources, so I guess they are not labelled with 14C. So how would one determine how much 14C from the biosolid has been incorporated into microbial biomass? Or is this using the natural abundance of 14C? but 14C as natural abundance is only useful to date centennial or millennial C, not the incorporation of new inputs into microbes which takes place over a few days to months…**

**Response:** Many thanks for your comments. We acknowledge that the description of the methods may have been confusing. To clarify, the monitored $^{14}C$ in this study originated from the added $^{14}C$-glucose, as stated in line 113 and described in Jones et al. (2018, 2019). The paragraph in question was intended as a disclaimer for readers, explaining the variation in the monitoring period of $^{14}CO_2$ release across temperatures (soil with bioamendments or mineral fertilizer). Since this is limited to both experiments 1 and 3, we initially placed it at the beginning of the methods. However, we have now moved this paragraph to the end of the methods section to improve the logical flow for readers.

*"In our incubation experiments, microbial C uptake is defined as the total labelled C remaining in the system, which has not been respired as $^{14}CO_2$ or incorporated in the microbial biomass (Glanville et al., 2016; Sánchez-Rodríguez et al., 2024). Notably, since there is no universally accepted protocol for soil incubation experiments exploring extreme heat-stress in the presence or absence of bioamendments (Schroeder et al., 2021), the duration of the heat-stress events conducted in this study (to assess soil microbes response in Experiments 1 and 3) was designed to reflect the typical duration and intensity of heatwaves experienced in the region where the soil samples were collected, which can last over a week with daily air temperatures reaching up to 45.4 °C (see Fig.*

*S1). A mechanistic approach was utilised in our experiments, where the soils were maintained at high temperatures for one week to explore microbial responses under an extreme, worst-case scenario, in Experiments 1 and 3. In Experiments 1 and 3, the monitorization period (14CO2 measurements) was extended until $^{14}CO_2$ was not detectable in the NaOH traps (Experiment 1: 27-days under 20 °C and 30 °C and 21 days under 40 °C and 50 °C samples; Experiment 3: 16 days) because the timing of 14CO2 emissions stabilisation was strongly influenced by temperature, which explains the differences in duration between experiments and across the different temperatures. The $^{14}CO_2$ monitoring used technique captures $^{14}CO_2$ emissions from both catabolic (i.e., rapid mineralisation) and anabolic (i.e., slow $^{14}CO_2$ release due to cell turnover) processes."*

Additionally, we have modified the last paragraph of the introduction section to make it clearer for readers according to this comment from Reviewer#2:

*"This study investigated the effects of a selection of bioamendments (composted olive mill pomace, composted biosolids, and composted solid urban residue) and a mineral fertiliser (diammonium phosphate) on microbial CUE and key soil chemical properties (including pH, labile C, N and available P) in two soils exhibiting contrasting carbonate and clay contents with a low P availability, a calcareous Vertisol and a non-calcareous Inceptisol, both collected from Mediterranean regions. We hypothesise that i) bioamendments will increase the availability of P and other nutrients supporting a more resilient soil microbial community with enhanced resistance to extreme heat-stress events than soils receiving mineral fertiliser or control soil with no P supply; ii) the enhancement of soil heat resistance will then increase microbial CUE in soils supplied with bioamendments; and iii) the calcareous Vertisol may exhibit a greater thermal and chemical buffering capacity under extreme heat events, supporting microbial metabolism at elevated temperatures more effectively than the non-calcareous Inceptisol, due to its higher pH and clay content, which help retain moisture. For that, three incubation experiments under controlled conditions were developed. Each experiment was designed to investigate different effects of extreme heat waves on soil microbes and functionality. In all cases, both soils were incubated at four different temperatures (20, 30, 40 and 50 °C). In the first experiment, soil respiration (measured as soil $^{14}CO_2$ emitted from soil following the addition of $^{14}C$-labelled glucose) was monitored during and after a heat stress event and microbial CUE calculated for the different soil × treatment (control, mineral fertiliser and bioamendment) × temperature combinations, while the second experiment (same experimental design) was focused on the effects on soil chemical properties (without $^{14}C$-labelled glucose application). A third experiment was conducted to assess the legacy effect of extreme heat-stress events on microbial activity in unamended soils, monitoring microbial activity after a heat stress when $^{14}C$-labelled glucose was added."*

**Reviewer: L202: what "monitoring period"? What "each experiment"? does this refer to each treatment (combination soil type/amendment)? Or each of the experiments numbered later?**

**Response:** We appreciate your comments; we have clarified this section and moved this part just before statistical analysis according to your comments (once we have explained Experiment 1, 2 and 3). Please, see our modified paragraph in the previous comment.

**Reviewer: L209: I am confused here: n=5, but further up (line 143): n=4. From the 4 times 100g prepared for each combination of bioamendment/soil type (40 pots in total: 2 soils x 5 bioamendments including no addition x 4 reps), line 143, how do we get to 5 replicates of bioamendment/soil type/temperature combinations?**

**Response:** Many thanks for your comment. We understand that this section could be confusing. We have simplified and modified this section to be improved accordingly:

*"Mixtures of the two soil types (Vertisol or Inceptisol) were prepared to explore the effect of mineral fertiliser vs bioamendments on microbial activity (Experiment 1) and soil nutrient cycling (Experiment 2) as a function of a simulated heat stress (20, 30, 40 and 50 °C). Consequently, the soils received varying quantities of mineral fertiliser or bioamendment, according to their P content (Table 2) to reach the target P level of 50 mg kg⁻¹, with a control treatment without mineral fertiliser or bioamendment. To ensure homogenisation after treatment (mineral fertiliser or bioamendment) application, larger mixtures of soil were prepared, because the experimental unit included only 2.5 g of soil. Therefore, soil was gradually added to pre-weighed fertiliser or bioamendment in larger batches, followed by thorough mixing to ensure homogeneity. Previously, the fertiliser and the bioamendments were ground and sieved to 0.5 mm to ensure homogeneity. From each homogenised batch, subsamples of each mixture (fertiliser / bioamendment and soil) were used for incubation experiments. In Experiment 3, soil (Vertisol or Inceptisol) was not mixed with any mineral fertiliser or bioamendment as we only evaluated the legacy effect of the heat stress (20, 30, 40 and 50 °C)."*

**Reviewer: L214: now I get it! 14C labelled glucose… so a glucose incorporation method is used as a standardised assay to quantify CUE.**

**Response:** Exactly, in this work (as stated in the aims of the manuscripts and the methods) we used labelled $^{14}$C glucose to monitor soil respiration in response to the heat stress in soils that were mixed (Experiment 1) or not (Experiment 3) with mineral fertilisers or bioamendments. This method is widely used to detect the shifts that could happens in microbial communities subjected to different external factors. The Introduction and Material and Material and Methods sections were deeply improved following Reviewer#2 comments and suggestions as can be seen in our previous responses and also here:

Introduction section: *"For that, three incubation experiments under controlled conditions were developed. Each experiment was designed to investigate different effects of extreme*

heat waves on soil microbes and functionality. In all cases, both soils were incubated at four different temperatures (20, 30, 40 and 50 ℃). In the first experiment, soil respiration (measured as soil $^{14}CO_2$ emitted from soil following the addition of $^{14}C$-labelled glucose) was monitored during and after a heat stress event and microbial CUE calculated for the different soil × treatment (control, mineral fertiliser and bioamendment) × temperature combinations, while the second experiment (same experimental design) was focused on the effects on soil chemical properties (without $^{14}C$-labelled glucose application). A third experiment was conducted to assess the legacy effect of extreme heat-stress events on microbial activity in unamended soils, monitoring microbial activity after a heat stress when $^{14}C$-labelled glucose was added."

In material and methods section:

*Experiment 1: Microbial activity during and after an extreme heat-stress event*
*Microbial activity during and after an extreme heat-stress event was assessed by measuring microbial 14CO2 release and CUE, following the methods described in Jones et al. (2019, 2018) after the incorporation of 14C-labelled glucose in soils receiving inorganic fertiliser, bioamendments, or no addition (control; see Fig. S4 for a schematic overview of the experimental design)."*

*"Experiment 3: "To understand the legacy effect of the heat-stress, soil respiration was monitored after the heat stress (during 16 days) and microbial CUE calculated at the end of the monitoring period in another incubation experiment. Briefly, 2.5 g of soil (n = 5 per combination of soil type and temperature; control soil only without any mineral fertiliser or bioamendment) was placed in a sterile 50 ml polypropylene centrifuge tube, wetted with 200 µl of DI H2O, and pre-incubated for 1-week at 20 ℃. After this week, soils were then placed in an incubator at 20 °C, 30 °C, 40 °C or 50 °C for another week. Then, soil samples were returned to 20 °C and 250 µl of 14C-labelled glucose (4.6 kBq ml-1, 10 mM; American Radiolabelled Chemicals Inc., St Louis, USA) was pipetted evenly onto the soil surface (see Fig. S4 Experiment 3 for an overview of the experimental design). NaOH traps were placed above the soil surface and changed on days 0.04, 0.13, 0.33, 1, 2, 3, 6, 8, 15 and 16, prior to measure 14CO2 via liquid scintillation counting. After 16 days, soil was extracted with fridge-cold 1 M NaCl to determine the amount of 14C remaining in the soil and microbial CUE calculated as described previously in Experiment 1."*

Additionally, we have added a new figure (Fig. S4) including detailed information of each incubation experiment, summarising the experimental design in each case (Experiment 1, 2 and 3). We hope that this figure helps the readers understand our study and improve our manuscript.

[Figure]

**Fig. S4.** Schematic overview of the experimental design. Experiment 1: Microbial activity during and after the heat stress (20, 30, 40 and 50 °C) was applied to the soils with the different treatments (control, inorganic fertiliser, or bioamendments). Experiment 2: Soil chemical properties (soil pH, EC, moisture, total N and C, ammonium ($NH_4^+$), nitrate ($NO_3^-$), and Olsen-P) as a function of the soil and treatment (control, inorganic fertiliser, or bioamendments) after the heat stress (20, 30, 40 and 50 °C). Experiment 3: The legacy effect of heat stress was evaluated in unamended soils after the heat stress (20, 30, 40 and 50 °C), followed by [14]C-glucose addition.

**Reviewer: L225: so when is incorporation into microbial biomass quantified? I suppose the extraction with NaCl extracts what is not in microbial biomass or respired? So one would need to deduce that from what is added originally and what is recovered as 14CO2 cumulatively to deduce what is in microbial biomass?**

**Response:** We appreciate this comment. The method that we use in our study to calculate microbial CUE, following Jones et al. (2019, 2018) and Sánchez-Rodríguez et al. (2024), estimates microbial immobilisation by using the following equation:

*"Additionally, microbial immobilisation of the [14]C-substrate ($^{14}C_{imm}$) after the monitoring period was estimated as follows:*

$$^{14}C_{imm} = {}^{14}C_{tot} - {}^{14}C_{NaCl} - {}^{14}CO_{0-t\ days} \qquad (1)$$

As Reviewer#2 mentioned, the extraction with 1 M NaCl is used to quantify the amount of [14]C recovered from the soil at the end of the experiments.

**Reviewer: L239: In experiment 1, 7 days at the different temperatures, including only 5 in the presence of 14C glucose. Ok; but why 9 days in experiment 2?**

**Response:** As Reviewer#2 commented, the incubation time in Experiment 1 and 3 is different from that used in Experiment 2. For clarification, all the samples of the three-experiment passed through 7 days of preincubation at 20 ºC and then, they were incubated at different temperatures.

It is important to realise that the three experiments have different objectives (see Material and Methods, in the description of each experiment, separately). Experiment 1 and 3 aimed to monitor the changes in soil respiration during and after the heat stress in both soils (Vertisol and Inceptisol) unamended (Experiment 3) and amended with the mineral fertiliser or bioamendments (Experiment 1), and after the heat stress in Experiment 3

Experiment 2 was focused on evaluating modifications in soil chemical properties after the heat stress. For that, the duration of the heat stress do not have to be exactly the same as in Experiment 1, and we decided to increase this time in two days (so, the heat wave in this case was simulated for 9 days; still within the values observed in our climatological data).

In conclusion, the methods were adjusted depending on the planned objectives in each experiment.

Please check our new version of the Introduction (last paragraph) and Material and methods where these changes were done according to this and other comments (see our responses to other suggestions to avoid repetition here) – this information was added in a previous response, so, to avoid repetition we have not included it again here.

**Reviewer: L254: and now it's a week of heating. But how long are soils at 20oC before 14C glucose addition?**

**Response:** Many thanks for your comment. We think this section is clearer now thanks to that. Please see our schematic overview of the experimental design, now included as a new figure in supplementary material (Fig. S4).

Summarizing, the samples (Experiments 1, 2 and 3) were wetted and preincubated for 1 week at 20 ºC before the incubation at different temperatures. Experiments 1 and 3 used labelled $^{14}$C glucose but it was not used in Experiment 2. In Experiment 1, labelled glucose was used with the main aim to detect the effect of heat stress during and after heat stress in soils previously amended with the different treatments; the samples were incubated for two days with different temperatures before the addition of $^{14}$C-labelled glucose and after that the incubation was prolonged to 7 days. In Experiment 2, the samples (mixed with bioamendments or mineral fertiliser) were heated for 9 days to reproduce extreme field conditions (like those observed under real conditions) without any supply of $^{14}$C-labelled glucose, as the objective was to evaluate alterations in soil chemical properties. Then, in Experiment 3, we used labelled glucose with the aim to detect the legacy effect of heat stress on unamended soils (after the heat stress); the soil samples were incubated at different temperatures for 7 days to be returned to 20ºC and

then [14]C-labelled glucose was added to measure microbes' respiration and calculate microbial CUE (at the end of the experiments).

**Reviewer: L256-257: as I suspected line 224-225, microbial biomass C of microbial biomass 14C were never measured, only the remaining 14C in the total soil is quantified. Explanations about how this is used to calculate CUE are needed! It says here "As described above" but I can't find the calculations/equations anywhere.**

**Response:** We appreciate this comment. As we previously mentioned, there are various methods that are used to estimate the amount of C that remains in the soil, however, in this research we have used the method described by Jones et al. (2019, 2018) to quantify the CUE, using the formulas:

*"Additionally, microbial immobilisation of the $^{14}C$-substrate ($^{14}C_{imm}$) after the monitoring period was estimated as follows:*

$$^{14}C_{imm} = {}^{14}C_{tot} - {}^{14}C_{NaCl} - {}^{14}CO_{0-t\ days} \qquad (1)$$

*where $^{14}C_{tot}$ is the total amount of $^{14}C$-substrate added to the soil, $^{14}C_{NaCl}$ is the amount of $^{14}C$ recovered from the soil in the 1 M NaCl extracts at the end of the experiments and $^{14}CO_{0-t\ days}$ is the total amount of $^{14}C$ recovered as $^{14}CO_2$ during the experiments (21-27 days). Then, microbial CUE for the $^{14}C$ substrate was estimated as follows following* *Jones et al. (2018a,b)*:

$$microbial\ CUE = {}^{14}C_{imm} / ({}^{14}C_{imm} + {}^{14}CO_{0-t\ days}) \qquad (2)"$$

**Reviewer: L258-260: Now I am a bit confused. L256, it is implied that all treatments are incubated for 16 days before extraction for remaining 14C. For experiment 1, lines 217-220, it is indeed indicated that incubation time difference between temperature treatments. So this seems to apply only to experiment 1. I think diverging incubation time for calculating CUE based on glucose incorporation are hugely problematic, as described in the general comment, and I question the validity of the approach to conclude anything g about the effect of temperature on CUE.**

**Response:** Many thanks for your comment. As described for each experiment (and included in previous responses), the monitoring of $^{14}CO_2$ was stopped once the rate of $^{14}CO_2$ had plateaued indicating the all the glucose added was mineralised. At that point, microbial CUE was evaluated. This approach allows us to capture the effect of the heat stress on microbial CUE before microbial adaptation and to minimize CUE overestimation. Since each temperature has a different impact on microbial activity—and higher temperatures accelerate glucose consumption and C sequestration—it seems more logical to adapt the monitoring $CO_2$ period to these slight differences (just a few days) for the different temperatures and calculate microbial CUE immediately at this point. This method is commonly used in similar studies to evaluate the impact of different treatments

on soil C, respiration and CUE (please, see the references that we have added in our previous responses dealing with this issue).